# Evaluation of Immunogenicity of *Mycobacterium tuberculosis ag85ab* DNA Vaccine Delivered by Pulmonary Administration

**DOI:** 10.3390/vaccines13050442

**Published:** 2025-04-23

**Authors:** Haimei Zhao, Zhen Zhang, Yong Xue, Nan Wang, Yinping Liu, Xihui Ma, Lan Wang, Xiaoou Wang, Danyang Zhang, Junxian Zhang, Xueqiong Wu, Yan Liang

**Affiliations:** 1Beijing Key Laboratory of New Techniques of Tuberculosis Diagnosis and Treatment, Institute of Tuberculosis Research, Senior Department of Tuberculosis, The Eighth Medical Center of PLA General Hospital, 100091 Beijing, China; zhm19960605@163.com (H.Z.); xueyong20170701@163.com (Y.X.); nanwang2015@163.com (N.W.); liuyinpingguo@126.com (Y.L.); maxihui@sina.com (X.M.); wanglan922@163.com (L.W.); wangxiaoou0113@163.com (X.W.); zdy956557865@163.com (D.Z.); zhangjunx309@163.com (J.Z.); 2Graduate School, Hebei North University, Zhangjiakou 075000, China; zhen090323@126.com

**Keywords:** *Mycobacterium tuberculosis*, DNA vaccine, pulmonary delivery, electroporation, immunogenicity

## Abstract

**Background:** Tuberculosis (TB) is a respiratory infectious disease, and the current TB vaccine has low local lung protection. We aim to optimize immune pathways to improve the immunogenicity of vaccines. **Methods:** In the immunogenicity study, 50 BALB/c mice were randomly divided into the following: (1) phosphate buffered saline (PBS)+intramuscular injection combined with electroporation (EP) group (100 μL), (2) pVAX1+EP group (50 μg/100 μL), (3) *ag85ab*+EP group (50 μg/100 μL), (4) pVAX1+pulmonary delivery (PD) group (50 μg/50 μL), and (5) *ag85ab*+PD group (50 μg/50 μL). Immunization was given once every 2 weeks for a total of three times. The number of IFN-γ-secreting lung and spleen lymphocytes was determined by enzyme-linked immunospot assay (ELISPOT). The levels of Th1, Th2, and Th17 cytokines in the culture supernatants of lung and spleen lymphocytes were detected with the Luminex method. The proportion of FoxP3 regulatory T cells in splenocytes was determined by flow cytometry. The levels of IgG-, IgG1-, and IgG2a-specific antibodies in plasma and IgA antibody in bronchoalveolar lavage fluid (BALF) were determined by enzyme-linked immunosorbent assay (ELISA). **Results:** The PD and EP routes of *Mycobacterium tuberculosis* (*M. tb*) *ag85ab* DNA vaccine can effectively induce the responses of IFN-γ-secreting lung and spleen lymphocytes, and induce dominant Th1 and Th17 cell immune responses. The PD route can induce earlier, greater numbers and stronger responses of pulmonary effector T cells, with higher levels of the specific antibody IgA detected in BALF. High levels of the specific antibodies IgG, IgG1, and IgG2α were detected in the plasma of mice immunized by the EP route. **Conclusions:** The PD route of DNA vaccines can more effectively stimulate the body to produce strong cellular and mucosal immunity than the EP route, especially local cellular immunity in the lungs, which can provide early protection for the lungs. It can significantly improve the immunogenicity of the *ag85ab* DNA vaccine, suggesting a feasible and effective approach to DNA immunization.

## 1. Introduction

*Mycobacterium tuberculosis* (*M. tb*) is a highly infectious pathogen that poses a serious threat to human health. The latest World Health Organization (WHO) report indicates that there were 10.8 million new cases of tuberculosis (TB) and 1.25 million deaths worldwide in 2023 [1]. Furthermore, TB has been identified as a major cause of death in developing countries. The respiratory tract is the natural route of infection for TB, with pulmonary TB being the most common form. Vaccination remains an important tool in the fight against TB, but the current vaccines have limited efficacy, particularly against severe forms of the disease. However, Bacillus Calmette–Guerin (BCG) is the only globally licensed TB vaccine and is included in China’s national immunization program, given by intradermal injection at birth, and protects children against severe TB, but its protective effect generally lasts only 10–15 years, and the protective effect of the BCG vaccine in adults varies widely and is generally less effective [2]. In addition, the BCG vaccine provides limited protection against pulmonary TB in endemic areas [3]. Therefore, it is particularly important to develop a new safe and effective TB vaccine or a new route of immunization to improve vaccine efficacy.

Ag85A and Ag85B are the primary secreted proteins of the *M. tb* Ag85 complex. These proteins transfer and deposit trehalose into the mycobacterial cell wall through their mycolic acid transferase activity, thus playing a critical role in the biosynthesis of the mycobacterial cell wall. They show high immunogenicity and can be recognized by the host immune system. They have been shown to stimulate T- and B-cell responses in tuberculosis patients; to induce delayed-type hypersensitivity, protective immunity, and specific antibody responses in *M. tb*-infected guinea pigs; and to induce proliferation of peripheral blood mononuclear cells in most PPD-positive individuals and a few patients with active tuberculosis. In 1996, Huygen et al. [4] reported for the first time that immunization of mice with the gene encoding the *M. tb* Ag85 complex induced strong cellular and humoral immune responses against live *M. tb* and BCG. In 1998, Denis et al. [5] demonstrated that mice immunized with an Ag85A DNA vaccine exhibited stronger and more extensive CD4^+^ T cell responses and cytotoxic T-lymphocyte (CTL) activity. In a separate study, Ha et al. [6] used an Ag85A DNA vaccine to treat a mouse model of *M. tb* infection and found that the DNA vaccine prevented the onset of latent TB infection in mice and shortened the duration of conventional chemotherapy. Our research group also used an Ag85A DNA vaccine alone or in combination with chemotherapy to treat a mouse model of MDR-TB and also showed good efficacy, which significantly reduced pulmonary and splenic tuberculosis mycobacterial loads after two months of treatment [7,8]. Zhu et al. [9] used an Ag85B DNA vaccine to treat a mouse model of tuberculosis infection, and found that the vaccine could induce a Th1-type immune response, produce high levels of IFN-γ and TNF-α, and reduce in lung and spleen colony counts by 1.2 and 0.7 logs, respectively. Consequently, our research team used the coding genes of these two antigens to construct an *M. tb ag85ab* chimeric DNA vaccine, which was able to induce humoral and cellular immune responses by immunizing mice by intramuscular injection and showed strong immunogenicity and significant therapeutic efficacy in a mouse TB model [10]. At present, 45.5% of the 22 vaccines in clinical trials worldwide are based on Ag85A and/or Ag85B antigens. Therefore, Ag85A and Ag85B are the most promising candidate target antigens for new TB subunit vaccines.

Although DNA vaccines can induce strong immunogenicity in mice [11,12], their application in large animals and humans has shown a significant lack of immunogenicity [13,14]. Therefore, there is an urgent need to develop effective means of vaccine delivery to enhance the immunogenicity of TB DNA vaccines in order to improve the protective efficacy of TB DNA vaccines. Currently, the main immunization routes of TB vaccines include intradermal injection [15,16], subcutaneous injection [17], intramuscular injection [18,19], mucosal immunization [20,21], and intravenous injection [22]. Individuals with intradermal BCG vaccination showed a strong cellular immune response against *M. tb*, but the protective efficacy was moderate, and the protective effect against natural aerosol infection varied widely [23]. In an animal model of subcutaneous BCG inoculation, a recall response to *M. tb* aerosol challenge was not detected in the lungs until 13 days after exposure to *M. tb* aerosol, and this delayed response allowed *M. tb* to proliferate and spread within macrophages [24]. Electroporation (EP) technology can enhance the efficacy of DNA vaccines by increasing cellular uptake of plasmid DNA, promoting sustained antigen expression, inducing potent cellular immunity, and reducing dose dependence, making it a preferred strategy for vaccine development. Previously, our research team found that DNA immunization by EP improved the immunogenicity of low-dose DNA vaccines, reduced the amount of vaccine immunized and increased the immunotherapeutic efficacy of the vaccine by administering different doses of *ag85ab* DNA vaccine by intramuscular injection in combination with EP [25]. In recent years, the study of the mucosal immunization of vaccines has become a hotspot, and common methods include nasal droplets [26], oral administration [27], and pulmonary delivery [28]. Intranasal BCG vaccination can rapidly activate pulmonary macrophages so that *M. tb* is mainly confined to macrophages, and its protective efficacy is significantly higher than that of subcutaneous vaccination [28]. Hand-held pulmonary drug delivery devices can deliver vaccines directly to the lungs, enabling them to exert their effect at the site of infection. Early on, Bhaskar et al. compared subcutaneous and aerosol-delivered BCG immunization in mice. The results showed that lung CFUs in mice infected with *M. tb* were significantly lower after aerosol immunization [29]. Mucosal vaccines are more likely than parenteral vaccines to induce protective immune responses at the site of infection by targeting the respiratory mucosa and lungs [30]. Existing research has shown that a lung-delivered viral vector vaccine targeting TB can induce robust humoral and cellular immune responses, exhibiting long-term protective effects [31]. Recently, the safety and immunogenicity of mucosal immunization has also been demonstrated in a population receiving aerosolized BCG for the first time [32]. Currently, with the rapid advancement of nanocarrier technology, pulmonary delivery of lipid-based nucleic acid vaccines for TB has emerged as a new focus in the field [33]. Compared with traditional immunization, mucosal immunization of vaccines has many advantages: (1) The lungs are blessed with a large absorptive surface and are rich in a large number of capillaries so that the vaccine delivered to the lungs can be rapidly absorbed through the membrane-like alveolar epithelial cells and then circulated systemically along with the bloodstream, which can significantly reduce the dosage of TB vaccine and directly increase the utilization rate of the vaccine. (2) The lungs contain a large number of alveolar macrophages, antigen-presenting cells (APCs), B cells, and other immune cells, which can deliver the TB vaccine directly to the lungs invaded by *M. tb*, effectively activating the immune cells and inducing strong and effective immune protective responses in the body [34]. Therefore, to improve the immunogenicity of the vaccine and enhance the local immunity in the lungs, this study used a pulmonary delivery device to deliver the vaccine to the mouse lungs so that vaccine immunization would be closer to the natural infection route of *M. tb*. By analyzing and comparing the immunogenicity of the *M. tb ag85ab* DNA vaccine delivered by two routes of immunization [EP and pulmonary delivery (PD)], we studied the immunological characteristics of the *M. tb ag85ab* DNA vaccine delivered directly to the lungs of mice. This research will lay the foundation for the establishment of an effective DNA vaccine delivery system.

## 2. Materials and Methods

### 2.1. Ethical Statement

All animal experiments were approved by the Animal Ethics Committee of the Eighth Medical Centre of China PLA General Hospital (Ethics Batch No. 3092023122013257187; approval date is 4 December 2023). Mice were cared for according to the standards of the Regulations on the Management of Laboratory Animals issued by the State Scientific and Technological Commission of the People’s Republic of China.

### 2.2. Experimental Animals

SPF-grade female 18–20 g BALB/c mice, aged 6–8 weeks, were purchased from Beijing Vital River Laboratory Animal Technology Co., Ltd., Beijing, China [license number: SCXK (Beijing) 2021-0006]. They were maintained in the animal laboratory of the Eighth Medical Center of the PLA General Hospital, Beijing, China, and the experimental mice were fed according to the specifications.

### 2.3. M. tb ag85a/b Chimeric DNA Vaccine and Recombinant Ag85AB Chimeric Protein

The construction method of the *M. tb ag85a/b* chimeric DNA vaccine has been reported previously [10]. Briefly, the DNA encoding amino acids 125–282 of the *M. tb* H37Rv Ag85B protein were amplified by PCR using specific oligonucleotide primers containing the recognition site of the restriction enzyme Acc I. The purified PCR product was inserted into nucleotides 430–435 (the Acc I site) of the *ag85a* gene and then cloned into the eukaryotic expression vector pVAX1 to construct an *M. tb ag85a/b* chimeric DNA vaccine. In this study, the *M. tb ag85ab* DNA vaccine was produced and purified by Guangzhou Baiyunshan Baidi Biopharmaceutical Co., Ltd., Guangzhou, China. Ag85AB chimeric protein was prepared and purified by Beijing Qinbang Biotechnology Co., Ltd., Beijing, China.

### 2.4. Immunization Regimen of ag85ab DNA Vaccine

The flowchart of the comparative immunogenicity study of the two routes of *M. tb ag85ab* DNA vaccine by EP and PD is shown in Figure 1. Fifty BALB/c mice were randomly divided into five groups as follows: (1) phosphate buffered saline (PBS)+EP group (100 μL); (2) pVAX1+EP group (50 µg/100 μL); (3) *ag85ab*+EP group (50 µg/100 μL); (4) pVAX1+PD group (50 μg/50 μL); (5) *ag85ab*+PD group (50 μg/50 μL). The immunization was given once every 2 weeks for a total of 3 times. Mice were sacrificed at 6 and 12 weeks after the last immunization. To reduce the pain and discomfort of the animals, the mice were anesthetized by intraperitoneal injection of 140 μL of 1% pentobarbital (article number: WS20050411, manufacturer: Sinopharm Chemical Reagent Co., Ltd., Shanghai, China). After confirming that the depth of anesthesia was adequate, the mice were subjected to terminal blood sampling from the retro-orbital venous plexus and subsequently euthanized by cervical dislocation. The blood of the mice was collected using 4 mL blood collection vessels containing heparin/lithium anticoagulant, and the serum was centrifuged for use.

### 2.5. Determination of the Number of IFN-γ-Secreting Effector T-Lymphocyte Spots in Lung or Spleen Lymphocytes by ELISPOT Assay

Six or four mice were sacrificed at 6 or 12 weeks after the last immunization, respectively, and lung and spleen lymphocytes were isolated from each group. Unfortunately, the number of IFN-γ-secreting effector T-lymphocyte spots in the lungs at 12 weeks after the last immunization was lost due to operator error.

Mouse pulmonary lymphocytes or splenic lymphocytes were plated on the ELISPOT plate at 1 × 10^5^/mL or 3 × 10^6^/mL, respectively, and 100 μL of cells were added to each well. Subsequently, 50 μL of complete cell culture medium {90% RPMI-1640 + 10% fetal bovine serum (FBS)} as a negative control, 90 µg/mL PHA as a positive control, and 90 µg/mL Ag85AB recombinant protein were individually inoculated, and the ELISPOT culture plate was placed in a cell incubator containing 5% CO_2_ at 37 °C for 24 h. The mouse IFN-γ ELISPOT PLUS kit (code: 3321-4APT-2, batch: 85, Mabtech, Stockholm, Sweden) was then used to detect the number of IFN-γ-secreting effector T-lymphocyte spots according to the instructions provided. The culture plates were scanned using an enzyme-linked automated spot image analyzer (S5 versa, Cellular technology Ltd., Cleveland, OH, USA) and the results of the spot data were read. The mean number of spot-forming cells (SFCs) was calculated for each group, comparing the values for the stimulation wells and negative wells in each group.

### 2.6. Determination of Th1, Th2, and Th17 Cytokine Levels in Lung and Spleen Lymphocyte Culture Supernatants Using the Luminex Method

Six mice were sacrificed at 6 weeks after the last immunization. Pulmonary and splenic lymphocytes were isolated and seeded into 96-well cell culture plates at the concentrations of 1 × 10^5^ cells/mL and 5 × 10^5^ cells/mL, respectively, at 100 μL per well. Cells were then stimulated with three conditions: (1) 50 μL complete cell culture medium (90% RPMI-1640 + 10% FBS) as negative control; (2) 90 µg/mL PHA as positive control; and (3) 90 µg/mL Ag85AB recombinant protein as experimental treatment. The cell culture plate was placed in a humidified incubator at 37 °C with 5% CO_2_ for 48 h. After 48 h of culture, the splenic lymphocytes were placed in a 1.5 mL centrifuge tube and centrifuged at 5000 rpm for 3 min. The resulting supernatants were carefully collected and stored at −80 °C for subsequent analysis. The cytokine levels of Th1 {IFN-γ, interleukin (IL)-2, tumor necrosis factor (TNF)-α, granulocyte-macrophage colony-stimulating factor (GM-CSF), IL-12p70, IL-18, IL-27}, Th2 (IL-4, IL-5, IL-6, IL-9, IL-10, IL-13), and TH17 (IL-17, IL-22) in the lung and spleen lymphocyte culture supernatants were determined using the ProcartaPlex Mouse Th1/Th2/Th17 Cytokine Panel (17-Plex) (batch no.: 230746-004, lot number: EPX170-26087-901, Invitrogen, Vienna, Austria) according to the manufacturer’s procedures.

### 2.7. Proportion of FoxP3 Regulatory T Cells in Mouse Splenocytes Detected by Flow Cytometry

Six or four mice were sacrificed at 6 or 12 weeks after the last immunization, respectively, and the splenic lymphocytes from each group of mice were isolated to a concentration of 5 × 10^6^/mL. 100 μL of splenic cells, 2 μL of CD4-FITC monoclonal antibody (clone: RM4-5, Cat. No. 553046, Lot: 0076381, Invitrogen, Carlsbad, CA, USA), and 5 μL of CD25-APC monoclonal antibody (clone: PC61, Cat.No. 557192, Lot: 99301687, Invitrogen, Carlsbad, CA, USA) were added to a 12 × 75 mm^2^ FACS tube and mixed on a vortex shaker for 3 s and then placed at 4 °C protected from light for 30 min. The cells were washed with 2 mL pre-cooled PBS, centrifuged at 1200 rpm at 4 °C for 5 min, and the supernatant discarded. One mL of freshly prepared eBioscience Fixation/Permeabilization application solution (1:3 dilution) (REF: 00-5123-43, Lot: 2176735, Invitrogen, Carlsbad, CA, USA) was added to each tube, and mixed on a vortex shaker for 3 s. The tubes were kept at 4 °C and protected from light for 40 min. Two mL of Permeabilization Buffer application solution (REF: 00-8333-56, Lot: 2178649, Invitrogen, Carlsbad, CA, USA) was added and centrifuged at 1200 rpm for 5 min at 4 °C; the supernatant was discarded, and the procedure was repeated twice. Then, 2 μL of rat serum was added and incubated at room temperature in the dark for 15 min. Two isotype controls were added to the isotype tube, and 5 μL of FoxP3-PE was added to the sample tube for intracellular staining, and the tubes were thoroughly mixed by vortex shaking for 10 s and then incubated at 4 °C for 40 min in the dark. Then, 2 mL of permeabilization buffer application solution was added and centrifuged at 1200 rpm at 4 °C for 5 min. The supernatant was discarded and the tubes were washed twice, 2 mL of pre-cooled PBS was added to each tube, and then the tubes were centrifuged at 1200 rpm at 4 °C for 5 min. After aspirating the supernatant, 500 μL of PBS was pipetted and stored at 4 °C. Finally, a flow cytometer (FACS Aria II, BD, Piscataway, NJ, USA) was used for detection.

### 2.8. Detection of Antigen-Specific Antibodies IgG, IgG1, IgG2α in Mouse Plasma and IgA in Alveolar Lavage by ELISA Method

Six or four mice were sacrificed at 6 and 12 weeks after the last immunization, respectively, and the blood anticoagulated with lithium/heparin (specification: 4 mL, item no.: 20220323, Guangzhou Yangpu Medical Instrument Co., Ltd., Guangzhou, China) and alveolar lavage fluid were collected; the anticoagulated blood was centrifuged, and the plasma was separated. The specific antibodies IgG, IgG1, IgG2α in plasma and IgA in the alveolar lavage fluid were detected by the ELISA method. The specific antibody procedure was as follows: 100 µL of Ag85AB recombinant protein (working concentration: 5 µg/mL) was added to each well and allowed to stand overnight at 4 °C and then blocked with 5% albumin from chicken egg white (Cat. No. CA1411-100G, Lot: CA29132802, Coolaber Science Technology Co., Ltd., Beijing, China) in PBS at 37 °C for 3 h. One hundred µL of 0.1% bovine albumin V (Cat. No. A8020, Lot: 1112No51, Solarbio, Beijing, China) PBST-diluted plasma (dilution ratio: 1:100) or 100 µL of undiluted alveolar lavage fluid was added and incubated at 37 °C for 2 h. Then, 100 µL of goat anti-mouse IgG-HRP (dilution 1:100,000) (lot: GR3299987-7, item No. ab6789, Abcam, Cambridge, UK) or goat anti-mouse IgG1-HRP or goat anti-mouse IgG2α-HRP antibodies (dilution 1:10,000) or horseradish peroxidase-labeled goat anti-mouse IgA (dilution 1:10,000) (lot: GR3412375-5, Abcam, Cambridge, UK) was added and incubated at 37 °C for 1 h. Then, 100 µL of TMB substrate solution was added for 5 min at room temperature in the dark and the color development was stopped with 2 M sulfuric acid solution. Finally, the OD450 values of the antibodies in mouse plasma and alveolar lavage fluid were determined using a microtiter plate reader (Thermo Fisher Scientific, Shanghai, China).

### 2.9. Experimental Data Processing and Statistical Analysis

SAS 9.4 software was used to statistically analyze the data from this experiment, and data were expressed as mean and standard deviation. One-way analysis of variance or two-way analysis of variance was used for quantitative data conforming to the normal distribution, Dunnett’s test was used for multiple comparisons, and a non-parametric test was used for quantitative data not conforming to the normal distribution, and GraphPad Prism 9.0 software was used for plotting. *p* < 0.05 indicated that the difference in the experimental results was statistically significant.

## 3. Results

### 3.1. Detection of IFN-γ-Secreting T-Lymphocyte Spots in Lung and Spleen Lymphocytes by ELISPOT Assay

The number of IFN-γ-secreting pulmonary lymphocytes at 6 weeks after the last immunization is shown in Figure 2A. The number of IFN-γ-secreting pulmonary T lymphocytes was significantly higher in the *ag85ab*+PD and *ag85ab*+EP groups than that in the PBS+EP, pVAX1+EP, and pVAX1+PD groups (*p* < 0.0001). Notably, the number of IFN-γ-secreting pulmonary T lymphocytes in the *ag85ab* +PD group was 2-fold higher than in the *ag85ab*+EP group (*p* < 0.0001). The results suggest that the ability of the *M. tb ag85ab* DNA vaccine delivered by the PD route to induce IFN-γ secretion by pulmonary T lymphocytes was significantly greater than that of the EP immunization.

The results of the number of IFN-γ-secreting splenic lymphocyte spots are shown in Figure 2B,C. The number of IFN-γ-secreting splenic T lymphocytes in the *ag85ab*+PD and *ag85ab*+EP groups was significantly higher than that in the PBS+EP, pVAX1+EP, and pVAX1+PD groups at both 6 and 12 weeks after the last immunization (*p* < 0.0001), but the difference between the *ag85ab*+EP and *ag85ab*+PD groups was not statistically significant (*p* > 0.05). In addition, the number of splenic lymphocyte spots secreting IFN-γ was also significantly decreased in the *ag85ab*+EP and *ag85ab*+PD groups at 12 weeks after the last immunization compared to 6 weeks after the last immunization (*p* < 0.001, *p* < 0.01) (Figure 2D). The results indicate that the PD and EP routes of *M. tb ag85ab* DNA vaccine can effectively induce higher numbers of IFN-γ-secreting splenic T lymphocytes. However, the magnitude of this cellular immune response exhibited a time-dependent attenuation in both vaccinated groups.

### 3.2. Determination of Th1, Th2, and Th17 Cytokine Levels in Lung and Spleen Lymphocyte Culture Supernatants by the Luminex Method

#### 3.2.1. Th1, Th2, and Th17 Cytokine Levels in Lung Lymphocyte Culture Supernatants

The levels of Th1, Th2, and Th17 cytokines in the culture supernatants of mouse lung lymphocytes at 6 weeks after the last immunization are shown in Figure 3. Among the Th1 cytokines produced (Figure 3A–G), the levels of IFN-γ, IL-2, GM-CSF, active IL-12P70, IL-18, and IL-27 in the lung lymphocyte culture supernatants of the *ag85ab*+PD group were significantly higher than those of the PBS+EP group, the pVAX1+EP group, and the pVAX1+PD group (all *p* < 0.05). The levels of IL-2, IL-12P70, IL-18, and IL-27 in the *ag85ab*+EP group were significantly higher than those in the PBS+EP group and the pVAX1+PD group (*p* < 0.01, *p* < 0.001 or *p* < 0.0001). The levels of IFN-γ, IL-12P70, IL-18, and IL-27 levels in the *ag85ab*+PD group were significantly higher than those in the *ag85ab*+EP group (*p* < 0.001, *p* < 0.0001). Remarkably, IFN-γ levels in lung lymphocyte culture supernatants were 4 times higher in the *ag85ab*+PD group than in the *ag85ab*+EP group.

Among the Th2 cytokines produced (Figure 3H–M), the levels of IL-4, IL-5, IL-6, IL-9, IL-10, and IL-13 in the lung lymphocyte culture supernatants of the *ag85ab*+PD group were significantly higher than those of the PBS+EP group, and the pVAX1+PD group (*p* < 0.05, *p* < 0.01, *p* < 0.001 or *p* < 0.0001). The levels of IL-5, IL-6, and IL-13 in the lung lymphocyte culture supernatants of the *ag85ab*+EP group were significantly higher than those of the pVAX1+PD group (*p* < 0.05 or *p* < 0.01), and the levels of IL-9 and IL-10 in the *ag85ab*+EP group were also significantly higher than those of the pVAX1+EP group (*p* < 0.001 or *p* < 0.0001). The level of IL-10 in the *ag85ab*+PD group was significantly higher than that in the *ag85ab*+EP group (*p* < 0.001). The Th1/Th2 cytokine (IFN-γ/IL-4) ratio results are shown in Figure 3N, and the ratio of Th1/Th2 in the *ag85ab*+PD group was significantly higher than that in the other groups (*p* < 0.001 or *p* < 0.0001). Importantly, the ratio in the *ag85ab*+PD group was two times higher than that in the *ag85ab*+EP group.

Among the Th17 cytokines produced (Figure 3O,P), the levels of IL-17A and IL-22 in the pulmonary lymphocyte culture supernatants of the *ag85ab*+PD and *ag85ab*+EP groups were significantly higher than those in the PBS+EP, pVAX1+EP, and pVAX1+PD groups (*p* < 0.001 or *p* < 0.0001). There is no significant difference between IL-17A and IL-22 levels in the *ag85ab*+PD group and those in the *ag85ab*+EP group (*p* > 0.05).

#### 3.2.2. Th1, Th2, and Th17 Cytokine Levels in Splenic Lymphocyte Culture Supernatants

The levels of Th1, Th2, and Th17 cytokines in splenic lymphocyte culture supernatants from mice at 6 weeks after the last immunization are shown in Figure 4. Among the Th1 cytokines produced (Figure 4A–G), the levels of IFN-γ, IL-2, TNF-α, GM-CSF, IL-12P70, and IL-18 in the splenic lymphocyte culture supernatants of the *ag85ab*+PD group and the *ag85ab*+EP group were significantly higher than those of the PBS+EP group, the pVAX1+EP group, and the pVAX1+PD group (*p* < 0.05, *p* <0.01, *p* < 0.001 or *p* < 0.0001). Notably, the levels of IL-2 and TNF-α were significantly higher in the *ag85ab*+PD group than in the *ag85ab*+EP group (*p* < 0.05).

Among the Th2 cytokines produced (Figure 4H–M), the levels of IL-4, IL-5, IL-6, IL-10, and IL-13 in the splenic lymphocyte culture supernatants of the *ag85ab*+PD group and *ag85ab*+EP group were significantly higher than those in the PBS+EP group (*p* < 0.05, *p* < 0.01, *p* < 0.001, or *p* < 0.0001). The levels of IL-6 and IL-9 in the *ag85ab*+PD group were significantly higher than those in the PBS+EP group and the pVAX1+EP group (*p* < 0.05, *p* < 0.01, or *p* < 0.0001). Notably, the IL-6 level in the splenic lymphocyte culture supernatant of the *ag85ab*+PD group was significantly higher than that of the *ag85ab*+EP group (*p* < 0.05), whereas the level of IL-10 was significantly lower than that of the *ag85ab*+EP group (*p* < 0.0001). The results of Th1/Th2 cytokines (IFN-γ/IL-4) are shown in Figure 4N. The ratio of Th1/Th2 cytokines in the splenic lymphocyte culture supernatant of the *ag85ab*+EP group was significantly higher than that in the other groups (*p* < 0.01 or *p* < 0.0001). The ratio of Th1/Th2 cytokines in the *ag85ab*+PD group was also significantly higher than that in the PBS+EP group, pVAX1+EP group, and pVAX1+PD group (*p* < 0.01).

Among the Th17 cytokines produced (Figure 4O,P), the levels of IL-17A and IL-22 in the splenic lymphocyte culture supernatants of the *ag85ab*+PD group were significantly higher than those of the other groups (*p* < 0.01, *p* < 0.001, or *p* < 0.0001). The levels of IL-22 in the *ag85ab*+EP group were significantly higher than those in the PBS+EP group, pVAX1+EP group and pVAX1+PD group (*p* < 0.01, *p* < 0.001, or *p* < 0.0001). Notably, the levels of IL-17A and IL-22 were significantly higher in the *ag85ab*+PD group than in the *ag85ab*+EP group (*p* < 0.001 or *p* < 0.0001).

### 3.3. Proportions of FoxP3 Regulatory T Cells in Mouse Splenocytes Detected by Flow Cytometry

The proportions of FoxP3 regulatory T cells in mouse splenocytes are shown in Figure 5. At 6 weeks after the last immunization, the proportion of FoxP3 regulatory T cells in splenocytes of the *ag85ab*+PD group and *ag85ab*+EP group was significantly higher than that of the PBS+EP group or pVAX1+EP group (*p* < 0.05), and that in *ag85ab*+PD group was higher than that of the *ag85ab*+EP group, but there was no statistical significance (*p* > 0.05). At 12 weeks after the last immunization, the proportion of FoxP3 regulatory T cells in the splenocytes of the *ag85ab*+PD group and *ag85ab*+EP group was still significantly higher than that of the PBS+EP group and pVAX1+EP group (*p* < 0.01, *p* < 0.001, or *p* < 0.0001). At 6 and 12 weeks after the last immunization, there was no significant difference between the *ag85ab*+PD group and the *ag85ab*+EP group (*p* > 0.05). However, it is important to note that the results may not fully reflect the true biological differences due to inconsistent sample sizes between groups.

### 3.4. Detection of Specific Antibodies IgG, IgG1, and IgG2α in Mouse Plasma and IgA in Alveolar Lavage Fluid by ELISA

The levels of IgG, IgG1, and IgG2α in mouse plasma are shown in Figure 6. The levels of specific antibodies IgG, IgG1, and IgG2α produced by the *ag85ab*+EP group at 6 and 12 weeks after the last immunization were significantly higher than those of the other groups (*p* < 0.01, *p* < 0.001, or *p* < 0.0001). However, there was no statistical difference between the plasma IgG, IgG1, and IgG2α levels of the *ag85ab*+PD group and those of the negative control group (*p* > 0.05). There was no statistical difference between the two *ag85ab*+EP groups or the two *ag85ab*+PD groups (*p* > 0.05) when compared at 6 and 12 weeks after the last immunization (*p* > 0.05). These results demonstrate that the *ag85ab* DNA vaccine delivered by electroporation can induce systemic immunization in mice to produce specific IgG antibodies.

The levels of specific antibody IgA in the alveolar lavage fluid of the mice are shown in Figure 7. The *ag85ab*+PD group produced significantly higher levels of specific antibody IgA than the PBS+EP group (*p* < 0.05), *ag85ab*+EP group (*p* < 0.001), and pVAX1+PD group (*p* < 0.01) at 6 weeks after the last immunization (Figure 7A). The *ag85ab*+PD group produced significantly higher levels of specific antibody IgA than the PBS+EP group at 12 weeks after the last immunization (*p* < 0.05) (Figure 7B). There was no statistical difference between the two *ag85ab*+EP groups or the two *ag85ab*+PD groups (*p* > 0.05) when compared at 6 and 12 weeks after the last immunization (*p* > 0.05). (Figure 7C). The results showed that the pulmonary route of administration was effective in inducing a mucosal immune response.

## 4. Discussion

*M. tb* usually infects the lungs via the respiratory tract. It is extremely cunning, able to evade the body’s immune responses by various means and successfully parasitize in host macrophages [35]. Its unique cell wall composition allows it to resist the bactericidal effects of macrophages. In addition, *M. tb* has the ability to evade the host immune system by secreting specific proteins that inhibit the bactericidal function of macrophages. Following infection, *M. tb* also induces a strong inflammatory response, resulting in the release of large numbers of inflammatory cells and cytokines. However, this response is often insufficient to eradicate the pathogen and can lead to tissue damage. Furthermore, *M. tb* has the ability to enter a latent state in the host, where it can persist for long periods of time. In immunocompromised individuals, this latent pathogen can be reactivated and cause disease manifestations [36]. Selection of the appropriate vaccination method and route has been shown to be one of the important factors in inducing effective immune responses to TB vaccine [37]. It is imperative to emphasize the importance of targeting the pulmonary immune response in the development of vaccines against *M. tb*. The lung is the primary site of infection for *M. tb*, and vaccines targeting the lung can elicit a local immune response and enhance lung defensive. For example, studies have shown that inducing an immune response in the lung mucosa to produce specific IgA antibodies can prevent the attachment and invasion of pathogens at an early stage of invasion [38,39,40,41]. In addition, lung-targeted vaccines have been shown to promote the development of lung-resident memory T cells [42], which have the ability to persist in the lung for longer periods and trigger a rapid and effective immune response upon subsequent exposure to *M. tb*. TB vaccine candidates, such as PPE15-LMQ, have been shown to have the ability to elicit robust CD4+ T-cell responses in the lungs, even after systemic administration. These responses exhibit a resident memory phenotype, which is associated with protective immunity against TB. This local immune response not only provides prolonged protection but also mitigates immunopathological damage [36]. Many studies on mucosal immunity have shown that multifunctional Th1 cells, which produce high levels of cytokines locally in the lungs, are more effective in protecting against TB than the systemic immune responses exhibited in the spleen [43,44,45,46]. To enhance the local lung immunity, this study used a pulmonary delivery device for the first time to deliver a DNA vaccine into the lungs of mice and compared the characteristics of their systemic immunity and local lung immunity with DNA vaccine by intramuscular electroporation.

Effector CD4^+^ T cells can be divided into at least three subpopulations of Th1, Th2, and Th17 cells. Recent research has confirmed that intracellular pathogens can proliferate irregularly and intermittently induce Th1-type cellular immune responses, producing Th1-type cytokines that help eliminate intracellular pathogens. Among these, IFN-γ is the most important member and plays a crucial and central role in host defense against *M. tb* infection [47]. IFN-γ can effectively activate macrophages, synergistically promote the transformation of CD8 T lymphocytes into cytotoxic effector cells with IL-2, and inhibit the differentiation and function of Th2 cells and the expression and function of anti-inflammatory molecules, thereby enhancing phagocytic and bactericidal abilities and limiting inflammation-related tissue damage [48]. This study on the vaccine immunogenicity showed that both pulmonary delivery and intramuscular electroporation of the *M. tb ag85ab* DNA vaccine could effectively induce a significant IFN-γ production by effector T lymphocytes in the lungs and spleen. Notably, at 6 weeks after the last immunization, the number of IFN-γ-producing pulmonary T lymphocytes following pulmonary delivery was three times higher than that observed with intramuscular electroporation, and the level of IFN-γ produced by pulmonary delivery was also four times higher than that observed with intramuscular electroporation, while the splenic lymphocyte responses induced by the two immunization routes were similar. This indicates that DNA vaccine by pulmonary delivery induces a stronger early response and a greater number of IFN-γ-secreting lung T lymphocytes than intramuscular electroporation and is more advantageous for early induction of local anti-TB immunity in the lung. At 12 weeks after the last immunization, there was a downward trend in the number of IFN-γ-secreting lung and spleen lymphocytes from both immunization routes, but they still exhibited high immunogenicity. Wu M et al. used a mannosylated chitosan-based DNA vaccine for intranasal immunization. The results obtained demonstrated that the vaccine was able to effectively induce cellular immune responses in the lungs of mice, thereby enhancing protection against *M. tb* infection. [49]. These indicate that pulmonary delivery of the *ag85ab* DNA vaccine may be more effectively in inducing early release of the local anti-TB immune mediator IFN-γ in the lung.

Other Th1 cytokines (IL-2 and TNF-α) and atypical Th1 cytokines (GM-CSF, IL-12p70, IL-18, and IL-27) all support the crucial anti-TB role of the Th1-type immune response. IL-2 is an important cytokine that promotes T-cell proliferation and transformation and mediates immune activation and immune regulation in the body [50]. It can restore T-cell dysfunction caused by sustained stimulation of *M. tb* [51]. TNF-α limits the spread of *M. tb* and generates the mechanism of anti-TB effect by synergistically activating macrophages and granuloma formation [52]. GM-CSF can promote granuloma formation and enhance the host’s anti-TB immune responses by promoting macrophage M1 polarization, maturation, and activation [53]. IL-12 is an important inducer of Th1-type responses, and its active form IL-12p70 can promote the IFN-γ response, and enhance the anti-TB activity of macrophages and NK cells [54]. IL-18, with the synergistic assistance of IL-12 or IL-15, increases the expression of IL-18Rα, further promoting the production of IFN-γ by T cells and NK cells [55]. IL-18 induces T lymphocytes to polarize towards the Th1 phenotype and induces immune cells to produce various cytokines [56]. IL-27 has a dual effect on innate and adaptive immunity, which can enhance the anti-TB immune responses by regulating the differentiation of T cells and innate immune cells. It can also induce IL-10 to inhibit Th2 and Th17 cells, increase PD-L1 expression, and regulate the development or transformation of regulatory T cells (Tregs) [57]. This study showed that both the pulmonary delivery and intramuscular electroporation routes of the *ag85ab* DNA vaccine can induce high levels of typical and atypical Th1 cytokines (except for lung TNF-α and spleen IL-27) in lung and spleen lymphocytes at 6 weeks after the last immunization, with most lung Th1 cytokine levels significantly higher in the pulmonary delivery route than in the intramuscular electroporation route. This demonstrates that pulmonary delivery of DNA vaccine can early enhance local anti-TB Th1 immunity in the lung, providing an immune basis for clearance of *M. tb* in the lung. Both immunization routes do not induce high levels of TNF-α production in lung lymphocytes, thus avoiding the potential immunopathological changes caused by excessive TNF-α production [52]. The pulmonary delivery of the *ag85ab* DNA vaccine not only induces significantly higher levels of IL-2 and TNF-α in splenic lymphocytes compared to intramuscular electroporation, but the two immunization routes have little difference in inducing systemic anti-TB immunity.

Th2 cells mainly induce humoral immune responses and produce Th2-type cytokines (IL-4, IL-5, IL-6, IL-9, IL-10, and IL-13) that increase the chances of survival of *M. tb* by decreasing the body’s anti-TB immunity and promoting *M. tb* immune escape and latent infection, mainly by suppressing Th1 immune responses and macrophage function [58]. Among these, IL-6 is a pleiotropic cytokine whose involvement in the acute phase response enhances the early host immune response to *M. tb*, particularly by promoting T-cell activation and Th17-cell differentiation and assisting in the formation of more effective granulomas to limit the spread of *M. tb*, suggesting that IL-6 may be beneficial to some extent for anti-TB protection [59]. On the other hand, IL-6 contributes to B-cell differentiation and antibody production, which can weaken the Th1-type cellular immune response and lead to dysregulation of the immune response, inhibit macrophage autophagy, promote inflammation, and increase pathological inflammation, which is detrimental to anti-TB protection [60]. Although this study showed that the pulmonary delivery and intramuscular electroporation routes of the *ag85ab* DNA vaccine significantly increased Th2 cytokines in lung and splenic lymphocytes in mice 6 weeks after the last immunization, there was a significant increase in the ratio of Th1/Th2 cytokines (IFN-γ/IL-4), demonstrating that immunization with the DNA vaccine induces predominantly Th1-type cellular immune responses in the lung and spleen, which are more effective in controlling *M. tb* infection [61,62]. Intranasal immunization with other types of vaccines (such as Mycoplasma pneumoniae P1C DNA vaccine and SARS-CoV-2 vaccine) has also been shown to significantly induce Th1- and Th17-cell responses in both the lung and spleen, inducing both local immune responses in the lung and systemic immune responses [63,64].

Th17-type cells induce barrier tissue-specific responses, mainly producing IL-17A, IL-17F, and IL-22 cytokines. IL-17 is a class of pro-inflammatory factors produced mainly by activated CD4^+^ T lymphocytes, especially in the early stages of inflammation, acting on a wide range of cells and tissues. Khader SA et al. suggested that vaccination induces CD4^+^ T cells producing IFN-γ or IL-17, and this IL-17 secretion promotes the expression of various chemokines in the lung and the recruitment of IFN-γ-producing T cells, thereby inhibiting the growth of *M. tb* [65,66]. A systematic review by Li Q et al. [23] confirmed that IL-17, as an effector molecule similar to IFN-γ, protects humans from *M. tb* after BCG vaccination and *M. tb* infection, and the current evidence does not support that IL-17 would be an inducer of tissue damage in TB. Sun M et al. [67] found that the expression of the Th17 cell gene module was higher in BCG-protected macaques than in unprotected macaques, and the absolute counts of IFNγ^−^IL-2^−^IL-17^+^TNF^−^Th17 CD4^+^ cells were higher in BCG-protected macaques than in unprotected macaques. IL-22 is produced by innate and adaptive immune cells and is involved in mucosal immune defense in the early stages of infection, which can help to maintain the integrity of the mucosal barrier, prevent pathogen invasion, and protect tissues from excessive damage, thereby playing a protective role in respiratory infections [68]. IL-22 has now been shown to induce tissue cells to express acute inflammatory proteins, mucus-associated proteins, or antimicrobial peptides, which may act in synergy with other cytokines to inhibit *M. tb* growth [69]. In addition, IL-22 plays an important role in the regeneration and repair of epithelial cells, which is beneficial for the healing of TB lesions [68]. IL-22 has been shown to be one of the main markers of protection against bovine TB after BCG vaccination, and IL-22 may be used as one of the predictors of vaccine efficacy [70]. Therefore, the present study showed that both the pulmonary delivery and intramuscular electroporation routes of the *ag85ab* DNA vaccine induced high levels of Th17 cytokine production by pulmonary and splenic lymphocytes at 6 weeks after the last immunization, and in particular, the levels of IL-17A and IL-22 induced by the pulmonary delivery route were significantly higher than those induced by the intramuscular electroporation route, which further demonstrated that immunization with *ag85ab* DNA vaccine by both immunization routes induced an effective anti-TB immunity that helps to prevent the progression of *M. tb* infection.

FoxP3 regulatory T cells (expressing CD4 and CD25) are a class of CD4⁺ T-cell subpopulations with immunosuppressive functions that negatively regulate the immune response to *M. tb* infection [71]. They can significantly suppress the immune function of effector T cells, downregulate the immune response of Th1/Th17-type cells, and suppress the body’s immune response to *M. tb*, which, on the one hand, may limit excessive inflammation and thus avoid tissue damage, but, on the other hand, may also inhibit effective clearance of pathogens in favor of intracellular growth and latency of *M. tb* [72]. In this study, *M. tb ag85ab* DNA vaccine immunized by the pulmonary delivery route and intramuscular electroporation route led to an increase in the proportion of FoxP3 Tregs in mouse splenocytes 6 weeks after the last immunization, and the proportion of FoxP3 Tregs by the pulmonary delivery route was still significantly elevated 12 weeks after the final immunization. Fedatto PF et al. [73] immunized mice with hsp65 DNA vaccine or heterologous immunization using BCG as priming and DNA-hsp65 as boosting (BCG/DNA-hsp65) or BCG as priming and culture filtrate protein (CFP)-CpG as boosting (BCG/CFP-CpG), and all induced a significant increase in the proportion of CD4^+^Foxp3^+^ Tregs in the spleen, but heterologous immunization with BCG/DNA-hsp65 or BCG/CFP-CpG was more immunoprotective than DNA hsp65. This suggests that the vaccine induces Th1/Th17-type cellular immune responses, on the one hand, and increases the number of Foxp3 Tregs, on the other hand, to inhibit aberrant over-reactivity of Th1/Th17-type cellular immunity, which consistently recruits phagocytosis and causes immunopathological injury in the lung. Vaccine-induced CD4^+^ T cells and Tregs in appropriate proportions can balance anti-TB protection and lung injury [73].

Previous studies have shown that elevated plasma antibody concentrations contribute to the clearance of *M. tb* in the body, mediate anti-TB protection, and enhance humoral immunity [74]. The conditioning effect of serum anti-arabinomannan IgG after vaccination has been shown to significantly inhibit the intracellular growth of *M. tb* [75]. Furthermore, our previous studies have shown that intramuscular injection of *ag85a* and *ag85ab* DNA vaccines could all strongly induce the production of specific antibodies IgG, IgG1, and IgG2α [10,25]. In the present study, the *M. tb ag85ab* DNA vaccine immunized by intramuscular electroporation could induce mice to produce high levels of specific antibodies IgG, IgG1, and IgG2α at 6 and 12 weeks after the final immunization, whereas the vaccine immunized by pulmonary delivery only produced low levels of plasma-specific antibodies IgG, IgG1, and IgG2α. The IgG2α/IgG1 ratio was greater than 1 in all groups except the PBS group at 6 weeks after the final immunization, indicating that both routes of vaccine immunization induced mainly Th1-type cellular immunity. Pulmonary mucosal IgA is an antibody found mainly in the respiratory tract that plays an important role in preventing *M. tb* from invading and spreading in the lungs, thus helping to prevent TB [76]. IgA, a specific antibody, can elicit mucosal immunity via the pulmonary delivery route of TB vaccine [77]. Increasing experimental evidence has demonstrated that IgA secreted in the lung in response to TB vaccine can provide relevant immune protection against mouse TB by neutralizing *M. tb* [49,78]. In this study, the pulmonary delivery of the *M. tb ag85ab* DNA vaccine at 6 and 12 weeks after the final immunization could induce high levels of specific antibody IgA production in the alveolar lavage fluid of mice for a longer duration, suggesting that the pulmonary delivery route can induce a strong mucosal immune response. Several other DNA vaccines (such as SARS-CoV-2 spike protein-CpG oligonucleotide vaccine [79], H5N1 avian influenza or H1N1 2009 virus hemagglutinin DNA vaccine [80], or mannosylated chitosan-formulated DNA vaccine [49]) are effective in inducing high levels of IgA antibodies in the lungs of mice after intranasal immunization. This shows that the pulmonary delivery route of DNA vaccine is significantly superior to the intramuscular electroporation route in inducing respiratory mucosal immunity, whereas the intramuscular electroporation route is superior to the pulmonary delivery route in inducing the body to generate a humoral immune response.

## 5. Conclusions

In conclusion, *M. tb ag85ab* DNA vaccine delivered by both the pulmonary delivery route and the intramuscular electroporation route were able to effectively induce IFN-γ-secreting effector T-lymphocyte responses in the lung and spleen and induced a predominance of Th1- and Th17-type cellular immune responses. However, the DNA vaccine delivered by the pulmonary delivery route was able to induce more pulmonary effector T cells and a stronger response and induced higher levels of specific antibody IgA production in the respiratory mucosa, whereas the intramuscular electroporation route induced higher levels of specific antibodies IgG, IgG1, and IgG2α in plasma. Therefore, the DNA vaccine immunized by pulmonary delivery can more effectively stimulate the body to produce stronger cellular immunity and mucosal immunity than intramuscular electroporation, especially local cellular immunity in the lung, which can significantly enhance the immunogenicity of *M. tb ag85ab* DNA vaccine. In the future, the *M. tb* challenge trial will be used to further validate the efficacy of the pulmonary delivery of the *ag85ab* DNA vaccine.

## Figures and Tables

**Figure 1 vaccines-13-00442-f001:**
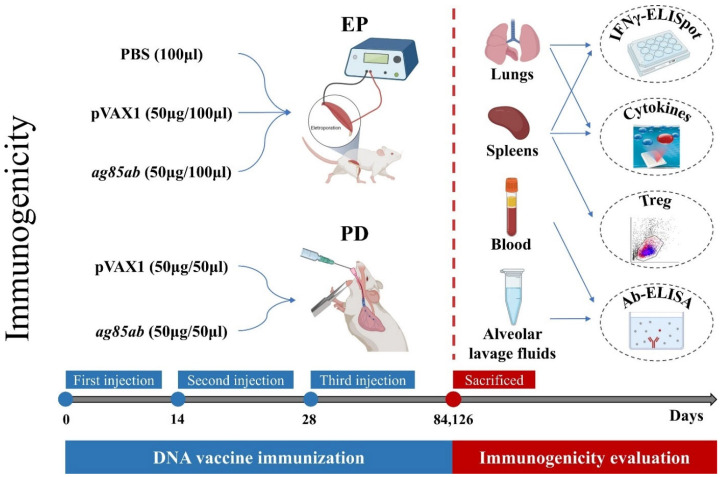
Flowchart of the immunogenicity experiment of the *M. tuberculosis ag85ab* DNA vaccine immunized by the pulmonary route. EP, electroporation; PD, pulmonary delivery.

**Figure 2 vaccines-13-00442-f002:**
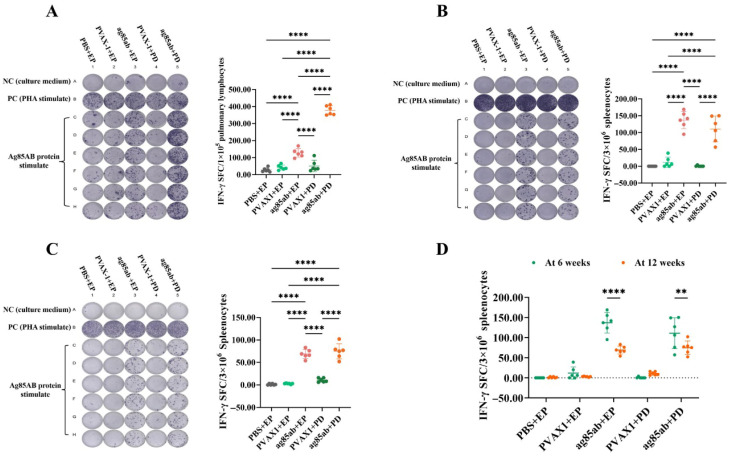
ELISPOT spot scan and the number of IFN-γ-secreting lung and spleen lymphocyte spots secreting IFN-γ in each group at 6 and 12 weeks after the last immunization. (**A**) Number of pulmonary lymphocyte spots at 6 weeks; (**B**) number of splenic lymphocyte spots at 6 weeks; (**C**) number of splenic lymphocyte spots at 12 weeks; (**D**) comparison of the number of splenic lymphocyte spots at 6 and 12 weeks after the last immunization. EP, electroporation; PD, pulmonary delivery. ** *p* < 0.01, **** *p* < 0.0001.

**Figure 3 vaccines-13-00442-f003:**
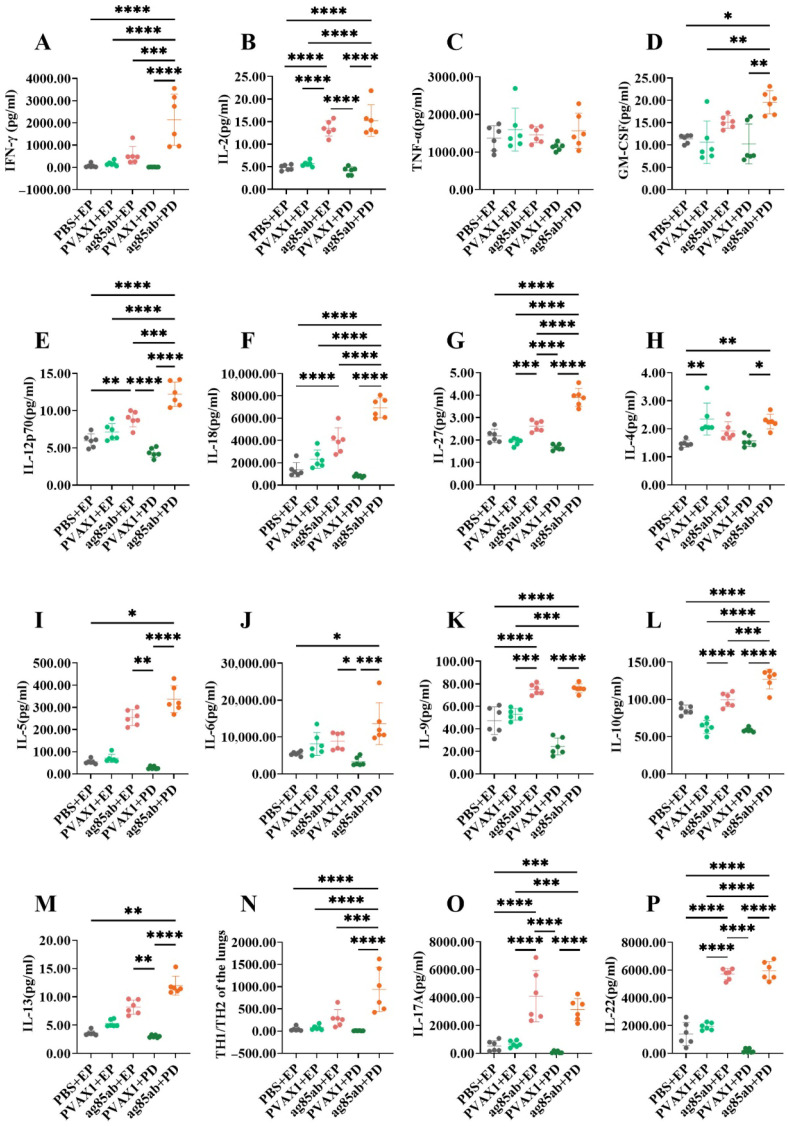
Th1, Th2, and Th17 cytokine levels in the lung lymphocyte culture supernatant of each group at 6 weeks after the last immunization. (**A**–**G**) Th1 cytokine levels (IFN-γ, IL-2, TNF-α, GM-CSF, IL-12p70, IL-18, and IL-27, respectively); (**H**–**M**) Th2 cytokine levels (IL-4, IL-5, IL-6, IL-9, IL-10, and IL-13, respectively); (**N**) Th1/Th2 (IFN-γ/IL-4) ratio; (**O**,**P**) Th17 cytokine levels (IL-17 and IL-22, respectively). EP, electroporation; PD, pulmonary delivery. * *p* < 0.05, ** *p* < 0.01, *** *p* < 0.001, **** *p* < 0.0001.

**Figure 4 vaccines-13-00442-f004:**
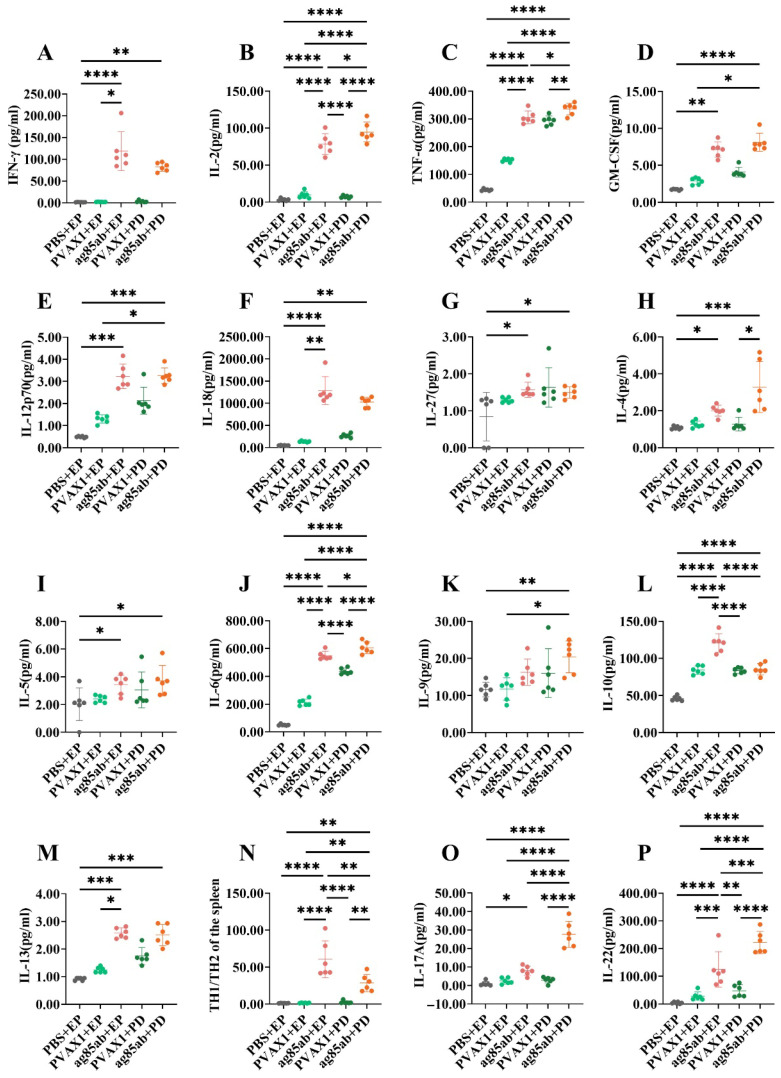
Th1, Th2, and Th17 cytokine levels in the spleen lymphocyte culture supernatant of each group at 6 weeks after the last immunization. (**A**–**G**) Th1 cytokine levels (IFN-γ, IL-2, TNF-α, GM-CSF, IL-12p70, IL-18, and IL-27, respectively); (**H**–**M**) Th2 cytokine levels (IL-4, IL-5, IL-6, IL-9, IL-10, and IL-13, respectively); (**N**) Th1/Th2 (IFN-γ/IL-4) ratio; (**O**–**P**) Th17 cytokine levels (IL-17 and IL-22, respectively). EP, electroporation; PD, pulmonary delivery. * *p* < 0.05, ** *p* < 0.01, *** *p* < 0.001, **** *p* < 0.0001.

**Figure 5 vaccines-13-00442-f005:**
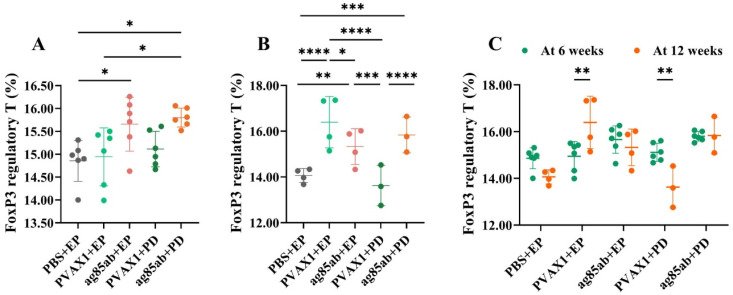
The proportion of FoxP3 regulatory T cells in the splenic lymphocytes of mice in each group at 6 (**A**) and 12 weeks (**B**) after the last immunization, and the comparison between 6 and 12 weeks (**C**). EP, electroporation; PD, pulmonary delivery. * *p* < 0.05, ** *p* < 0.01, *** *p* < 0.001, **** *p* < 0.0001.

**Figure 6 vaccines-13-00442-f006:**
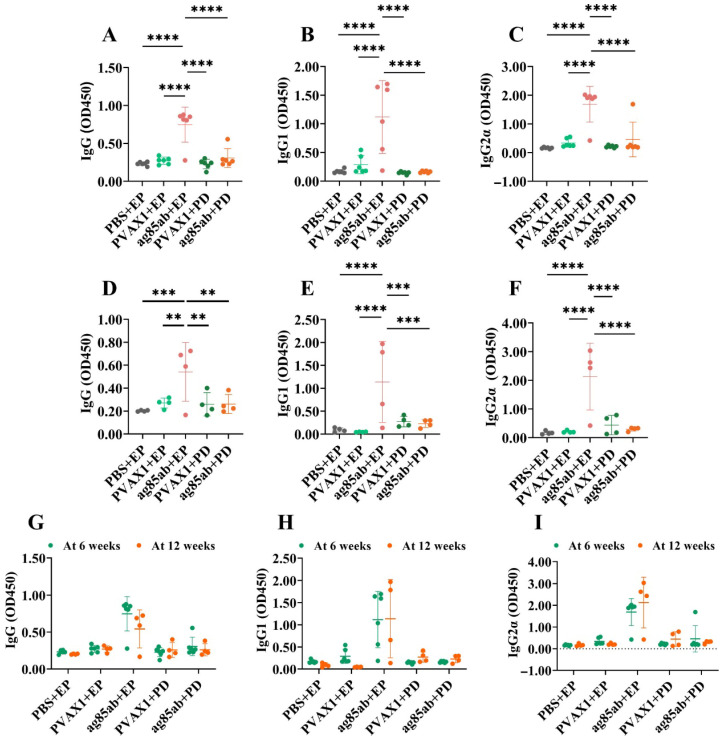
Plasma-specific antibody IgG, IgG1, and IgG2α levels in each group 6 and 12 weeks after the last vaccine immunization. (**A**–**C**) At 6 weeks, (**D**–**F**) at 12 weeks, (**G**–**I**) comparison of antibody levels between 6 weeks and 12 weeks after the last immunization. EP, electroporation; PD, pulmonary delivery. ** *p* < 0.01, *** *p* < 0.001, **** *p* < 0.0001.

**Figure 7 vaccines-13-00442-f007:**
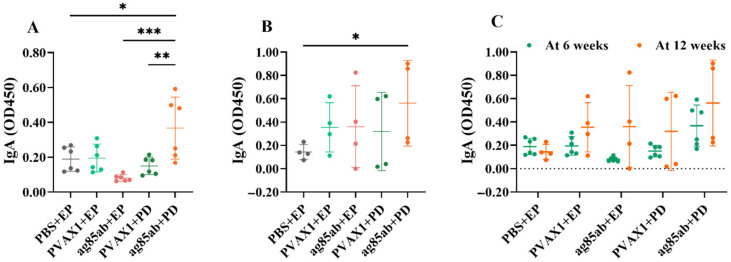
Specific antibody IgA levels in the mouse alveolar lavage fluid in each group at 6 (**A**) and 12 weeks (**B**) after the last immunization with the *ag85ab* DNA vaccine, and comparison of IgA levels between 6 weeks and 12 weeks after the last immunization (**C**). EP, electroporation; PD, pulmonary delivery. * *p* < 0.05, ** *p* < 0.01, *** *p* < 0.001.

## Data Availability

All data generated are presented in the manuscript.

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
