# Peer review of "Evaluation of Immunogenicity of Mycobacterium tuberculosis ag85ab DNA Vaccine Delivered by Pulmonary Administration"

_vaccines, 2025, doi:10.3390/vaccines13050442_

Round 1
Reviewer 1 Report
Comments and Suggestions for Authors
Introduction
Line 55: The statement "Vaccines are one of the most effective means of preventing TB" is potentially controversial. This is because there are limited vaccine options available for
tuberculosis (TB). It would be more accurate to note that, while vaccines like BCG provide some protection, the development of more effective TB vaccines remains a significant challenge. A clearer, more nuanced statement could be: "Vaccination remains an important tool in the fight against TB, but the current vaccines have limited effectiveness, particularly against severe forms of the disease."
Line 63: Justify the use of Ag85A and Ag85B as therapeutic targets and explain their significance in TB pathogenesis. What are their characteristics, and why are they relevant? Additionally, it’s necessary to better explain the characteristics observed when using electroporation with these antigens. Provide more in-depth discussion of the role of Ag85A and Ag85B in TB's virulence and pathogenesis to justify their selection as therapeutic targets. Clarify the relevance of electroporation as a delivery method for these antigens.
Methodology
Why were only female mice used? How could this affect the results? What justifies the use of only these mice?
References regarding infection via the tail with the BCG strain: Are there any precedents for this type of infection? More emphasis should be placed on this point.
Line 154: The number of groups used was five
Line 197; It is necessary to clarify under which conditions one stimulus or another was added. It is unclear if the induction was by Ag or PHA.
It is necessary to describe how the lymphocyte culture was carried out for supernatant collection. Additionally, an explanation should be provided as to why cytokines were not evaluated in situ, from cell lysates or plasma.
Why were the experiments conducted with M. bovis? It seems important to conduct experiments with a more representative strain of the infection, such as a virulent strain.
It would be useful to include a brief description of the vaccine creation process.
Results
Make sure the term is homogenized throughout the text to maintain consistency. Figure 1 (PXAX1/ PVAX1)
What tests were used? It is important to note that some conditions were performed with different sample sizes (n), which could bias the results (see figure 5).
Inconsistencies in the number of samples analyzed are observed (figure 7A), as the methodology mentions the sacrifice of 4 or 6 mice.
Author Response
We sincerely appreciate your positive evaluation of our manuscript and your recognition of the significance of our research. We are also very grateful for your valuable comments and suggestions. Your insightful review has greatly contributed to improving the clarity and scientific rigor of our work. We have carefully revised the manuscript to address your concerns. Below are our point-by-point responses to your suggestions.
Response to Reviewer 1’s comments:
Introduction:
- Comment: Line 55: The statement "Vaccines are one of the most effective means of preventing TB" is potentially controversial. This is because there are limited vaccine options available for tuberculosis (TB). It would be more accurate to note that, while vaccines like BCG provide some protection, the development of more effective TB vaccines remains a significant challenge. A clearer, more nuanced statement could be: "Vaccination remains an important tool in the fight against TB, but the current vaccines have limited effectiveness, particularly against severe forms of the disease."
Responses: Thank you very much for your valuable suggestion. I have changed "Vaccines are one of the most effective means of preventing TB" to "Vaccination remains an important tool in the fight against TB, but the current vaccines have limited effectiveness, particularly against severe forms of the disease."(Page 2, line 60)
- Comment: Line 63: Justify the use of Ag85A and Ag85B as therapeutic targets and explain their significance in TB pathogenesis. What are their characteristics, and why are they relevant? Additionally, it’s necessary to better explain the characteristics observed when using electroporation with these antigens. Provide more in-depth discussion of the role of Ag85A and Ag85B in TB's virulence and pathogenesis to justify their selection as therapeutic targets. Clarify the relevance of electroporation as a delivery method for these antigens.
Responses: Thank you very much for your valuable suggestion. In the introduction section, we have added the scientific rationale for Ag85A and Ag85B as therapeutic targets and provided a detailed explanation of their roles in the pathogenesis of tuberculosis (TB). Furthermore, we highlighted the advantages and suitability of electroporation for DNA vaccine delivery to provide additional support for the methodological choices in our study (Page 2, line 72). The revised content is presented below:
“Ag85A and Ag85B are the primary secreted proteins of the M. tb Ag85 complex. These proteins transfer and deposit trehalose into the mycobacterial cell wall through their mycolic acid transferase activity, thus playing a critical role in the biosynthesis of the mycobacterial cell wall. They are high immunogenicity and can be recognized by the host immune system. They have been shown to stimulate T and B cell responses in tuberculosis patients, to induce delayed-type hypersensitivity, protective immunity and specific antibody responses in M. tb-infected guinea pigs, and to induce proliferation of peripheral blood mononuclear cells in most PPD-positive individuals and a few patients with active tuberculosis. In 1996, Huygen et al. [4] reported for the first time that immunization of mice with the gene encoding the M. tb Ag85 complex induced strong cellular and humoral immune responses against live M. tb and BCG. In 1998, Denis et al. [5] demonstrated that mice immunized with an Ag85A DNA vaccine exhibited stronger and more extensive CD4+ T cell responses and cytotoxic T lymphocyte (CTL) activity. In a separate study, Ha et al [6] used an Ag85A DNA vaccine to treat a mouse model of M. tb infection and found that the DNA vaccine prevented the onset of latent TB infection in mice and shortened the duration of conventional chemotherapy. Our research group also used Ag85A DNA vaccine alone or in combination with chemotherapy to treat a mouse model of MDR-TB, and also showed good efficacy, which significantly reduced pulmonary and splenic tuberculosis mycobacterial loads after two months of treatment [7, 8]. Zhu et al. [9] used an Ag85B DNA vaccine to treat a mouse model of tuberculosis infection, and found that the vaccine could induce a Th1-type immune response, produce high levels of IFN-γ and TNF-α, and reduce in lung and spleen colony counts by 1.2 and 0.7 logs, respectively. Consequently, our research team used the coding genes of these two antigens to construct a M. tb ag85ab chimeric DNA vaccine, which was able to induce humoral and cellular immune responses by immunizing mice by intramuscular injection, and showed strong immunogenicity and significant therapeutic efficacy in a mouse TB model [10]. At present, 45.5% of the 22 vaccines in clinical trials worldwide are based on Ag85A and/or Ag85B anti-gens. Therefore, Ag85A and Ag85B are the most promising candidate target antigens for new TB subunit vaccines.
Although DNA vaccines can induce strong immunogenicity in mice [11, 12], their ap-plication in large animals and humans has shown a significant lack of immunogenicity [13, 14]. Therefore, there is an urgent need to develop effective means of vaccine delivery to enhance the immunogenicity of TB DNA vaccines in order to improve the protective efficacy of TB DNA vaccines. Currently, the main immunization routes of TB vaccines include intradermal injection [15, 16], subcutaneous injection [17], intramuscular injection [18, 19], mucosal immunization [20, 21], and intravenous injection [22]. Individuals with intra-dermal BCG vaccination showed a strong cellular immune response against M. tb, but the protective efficacy was moderate, and the protective effect against natural aerosol infection varied widely [23]. In an animal model of subcutaneous BCG inoculation, a re-call response to M. tb aerosol challenge was not detected in the lungs until 13 days after exposure to M. tb aerosol, and this delayed response allowed M. tb to proliferate and spread within macrophages [24]. Electroporation (EP) technology can enhance the efficacy of DNA vaccines by increasing cellular uptake of plasmid DNA, promoting sustained antigen expression, inducing potent cellular immunity, and reducing dose dependence, making it a preferred strategy for vaccine development. Previously, our research team found that DNA immunization by EP improved the immunogenicity of low-dose DNA vaccines, reduced the amount of vaccine immunized and increased the immunotherapeutic efficacy of the vaccine by administering different doses of ag85ab DNA vaccine by intramuscular injection in combination with EP [25].”
[4] Huygen K; Content J; Denis O; Montgomery DL; Yawman AM; Deck RR; DeWitt CM; Orme IM; Baldwin S; D'Souza C; Drowart A; Lozes E; Vandenbussche P; Van Vooren JP; Liu MA; Ulmer JB; Immunogenicity and protective efficacy of a tuberculosis DNA vaccine, Nat Med, 1996, 2: 857-859.
[5] Denis O; Tanghe A; Palfliet K; Jurion F; van den Berg TP; Vanonckelen A; Ooms J; Saman E; Ulmer JB; Content J; Huygen K; Vaccination with plasmid DNA encoding mycobacterial antigen 85A stimulates a CD4+ and CD8+ T cell epitopic repertoire broader than that stimulated by M. tuberculosis H37Rv infection, Infect Immun, 1998, 66: 1527-1533.
[6] Ha SJ; Jeon BY; Youn JI; Kim SC; Cho SN; Sung YC; Protective effect of DNA vaccine during chemotherapy on reactivation and reinfection of Mycobacterium tuberculosis, Gene Therapy, 2005, 12: 634-638.
[7] Liang Y; Wu X*; Zhang J; Li N; Yu Q; Yang Y; Bai X; Liu C; Shi Y; Liu Q; Zhang P; Li Z; The treatment of mice infected with multi-drug-resistant Mycobacterium tuberculosis using DNA vaccines or in combination with rifampin, Vaccine, 2008, 26(35): 4536-4540.
[8] Liang Y, Wu X*, Zhang J, Yang Y,Wang L, Bai X, Yu Q, Li N, Li Z. Treatment of multi-drug-resistant tuberculosis in mice with DNA vaccines alone or in combination with chemotherapeutic drugs, Scand J Immunol, 2011, 74(1): 42-46.
[9] Zhu D; Jiang S; Luo X; Therapeutic effects of Ag85B and MPT64 DNA vaccines in a murine model of Mycobacterium tuberculosis infection, Vaccine, 2005, 23(37): 4619-4624.
Methodology:
- Comment: Why were only female mice used? How could this affect the results? What justifies the use of only these mice?
Responses: Thank you very much for your valuable comments. In response to your question about the use of female BALB/c mice in tuberculosis vaccine research, we hereby explain the following reasons and their impact on the research results:
(1) BALB/c mouse is an inbred strain widely used in scientific research. It has a highly consistent genetic background, which means that the variability of the experimental results between different individuals is small, thus ensuring high repeatability of the experiment. Granuloma structure is not typical, tissue destruction is more serious, and survival is shortened [1,2]. Moreover, infection through either respiratory tract or intravenous injection can cause progressive pathological changes in the lungs of mice, thus successfully establishing a mouse tuberculosis model [1,3].
(2) BALB/c and C57BL/6 mice have strong resistance to MTB infection, but BALB/c mice have significantly weakened resistance to MTB after intravenous injection of large amounts of MTB. Compared with C57BL/6 mice, BALB/c mice usually had higher bacterial loads in the lung and spleen, and more extensive lung inflammation. Granuloma structure is not typical, tissue destruction is more severe, and survival is shortened [1,2]. Moreover, infection through either respiratory tract or intravenous injection can cause progressive pathological changes in the lungs of mice, thus successfully establishing a mouse tuberculosis model [1,3].
(3) The selection of mouse strain has a significant influence on vaccine evaluation. It has been found that pulmonary interstitial macrophages in TB resistant mice (such as BALB/c, C57BL/6 and B6) have a stronger ability to intracellular MTB than those in TB susceptible mice (CBA/J, C3H/HeJ, A/J, DBA/2). Moreover, TB-resistant mice are also better able to recruit T lymphocytes to the lungs after infection than TB-susceptible mice, and the activation of T cells in the lungs is more balanced [2,4,5]. In the immune memory animal model, histological studies confirmed that the number of memory T cells in the spleen of TB susceptible mice was significantly reduced, and the migration of T lymphocytes to the lungs was significantly reduced and delayed, which may be caused by insufficient up-regulated expression of adhesion and integrin molecules [5]. Moreover, the efficiency of evaluating candidate vaccines using mice with deficient memory immune cell response is not ideal [5]. Medina E and North RJ [6] used DBA/2 susceptible mice and BALB/c resistant mice to evaluate the anti-tuberculosis protective effect of BCG immunization, and the results showed that the vaccine had similar immune protective effect on the liver and spleen of the two kinds of mice, but the lung colony count and pathological changes of BALB/c mice were significantly better than those of DBA/2 susceptible mice. In addition, our study and others have shown that immunization of BALB/c mice with BCG or recombinant BCG and TB DNA vaccine can induce Th1-type immune responses [7-10]. These results indicate that BALB/c mice have significant advantages in immunity evaluation of tuberculosis vaccine.
(4) Female mice can be selected for the experiment to avoid the interference to the experimental results caused by gender differences and fluctuations in sex hormone levels, but in general, gender differences have little impact on the results, and the main purpose of maintaining gender consistency in the experiment is to reduce the difference. Therefore, the selection of female mice can ensure the stability and repeatability of the research results and improve the reliability of the experimental data.
In conclusion, the application of female BALB/c mice in this study helped us to more accurately evaluate the anti-TB protective effect of the vaccine.
[1] Medina, E., & North, R. J. (1998). Resistance ranking of some common inbred mouse strains to Mycobacterium tuberculosis and relationship to major histocompatibility complex haplotype and Nramp1 genotype. Immunology, 93(2), 270-274.
[2] Chackerian AA, Behar SM. Susceptibility to Mycobacterium tuberculosis: lessons from inbred strains of mice. Tuberculosis (Edinb). 2003;83(5):279-85.
[3] Dunn PL, North RJ. Virulence ranking of some Mycobacterium tuberculosis and Mycobacterium bovis strains according to their ability to multiply in the lungs, induce lung pathology, and cause mortality in mice. Infect Immun. 1995 Sep;63(9):3428-37.
[4] Lyadova IV, Eruslanov EB, Khaidukov SV, Yeremeev VV, Majorov KB, Pichugin AV, Nikonenko BV, Kondratieva TK, Apt AS. Comparative analysis of T lymphocytes recovered from the lungs of mice genetically susceptible, resistant, and hyperresistant to Mycobacterium tuberculosis-triggered disease. J Immunol. 2000 Nov 15;165(10):5921-31.
[5] Gruppo V, Turner OC, Orme IM, Turner J. Reduced up-regulation of memory and adhesion/integrin molecules in susceptible mice and poor expression of immunity to pulmonary tuberculosis. Microbiology (Reading). 2002 Oct;148(Pt 10):2959-2966.
[6] Medina E, North RJ. Genetically susceptible mice remain proportionally more susceptible to tuberculosis after vaccination. Immunology. 1999 Jan;96(1):16-21.
[7] Christy A. J., Dharman K., Dhandapaani G., et al. Epitope based recombinant BCG vaccine elicits specific Th1 polarized immune responses in BALB/c mice. Vaccine. 2012;30(7):1364–1370.
[8] Dobakhti F., Naghibi T., Taghikhani M., et al. Adjuvanticity effect of sodium alginate on subcutaneously injected BCG in BALB/c mice. Microbes and Infection. 2009;11(2):296–301.
[9] Liang Y, Wu X, Zhang J, et al. Immunogenicity and therapeutic effects of Ag85A/B chimeric DNA vaccine in mice infected with Mycobacterium tuberculosis [J]. FEMS Immunol Med Microbiol, 2012, 66(3): 419-26.
[10] Liang Y, Zhao Y, Bai X, et al. Immunotherapeutic effects of Mycobacterium tuberculosis rv3407 DNA vaccine in mice [J]. Autoimmunity, 2018, 51(8): 417-22.
- Comment: References regarding infection via the tail with the BCG strain: Are there any precedents for this type of infection? More emphasis should be placed on this point.
Responses: Thank you very much for your valuable suggestion. In this study, BCG strain was used to prepare a mouse tuberculosis model instead of Mycobacterium tuberculosis for the first time for preliminary evaluation of the new tuberculosis vaccine. The reasons are as follows: (1) Biological safety: BCG is a live attenuated vaccine strain of bovine Mycobacterium tuberculosis and has almost no pathogenic effect on humans (except immunocompromised individuals), while MTB is highly infectious and pathogenic and easily transmitted by aerosol. Therefore, the operation of Mycobacterium tuberculosis requires a biosafety level 3 laboratory (BSL-3), and the protection is costly and risky. However, BCG is a live attenuated vaccine strain of Mycobacterium bovis, which can be operated in biosafety level 2 laboratory (BSL-2), and the threat to laboratory personnel and the environment is significantly reduced [1]. (2) Feasibility of BCG infection model: MTB is highly pathogenic, and mice infected by intravenous injection are prone to severe disease and even death, which increases the risk and complexity of the experiment [2]. The virulence and pathogenicity of BCG strains are greatly reduced, and they can survive for weeks to months in mice, which makes it easier to establish a controllable mouse BCG infection model [2]. (3) Mouse BCG infection model to evaluate vaccine feasibility: This strategy is mainly based on the following rationale: (A) A TB vaccine that can successfully reduce MTB proliferation should also be able to reduce the replication of BCG strains. (B) Application of MTB to the human challenge model was not possible for safety and ethical reasons. (C) The key is that several studies have been reported on the use of BCG strain human challenge model to evaluate the effectiveness of TB vaccines faster and cheaper [3-5]. The results show that the BCG human challenge model can detect differences in anti-TB immunity induced by vaccination, thereby early, safe and rapid clinical evaluation of candidate TB vaccines and promoting vaccine progress. Krishnan N et al. [6] also successfully used a mouse BCG skin challenge model to evaluate tuberculosis vaccine efficacy. Therefore, in this study, it is feasible to use BCG strain instead of MTB to prepare a mouse tuberculosis model for preliminary screening of candidate vaccines, which can reduce the cost of early research and ensure that the most promising candidate vaccine is selected to further verify the preliminary results by using MTB infection animal models.
The intravenous route of infection was used in this study for the following reasons: (1) Both BALB/c and C57BL/6 mice were highly resistant to MTB infection, but the resistance of BALB/c mice to MTB infection was significantly weakened after intravenous injection of a large amount of MTB. The bacterial loads in the lung and spleen of BALB/c mice were usually higher than those of C57BL/6 mice, which accelerated the course of disease. Reduced survival [7,8]. (2) Compared with inhalation infection (depending on the depth of breathing of mice, there are some individual differences) or subcutaneous inoculation (local infection mainly), tail vein injection of BCG directly into the blood circulation leads to rapid spread of bacteria to systemic organs (such as lung, liver and spleen), mimicking hematogenous disseminated infection of MTB. The distribution of bacterial load in organs is more uniform and the experimental repeatability is higher. It is more suitable for evaluating the systemic immune protection and organ-specific protective effect of vaccines [9,10]. (3) The establishment of mouse tuberculosis model by tail vein injection has the advantages of simple operation, high infection efficiency, short experimental period and relatively low cost.
[1] National Health Commission of the People's Republic of China. List of pathogenic microorganisms that are transmitted between humans. 2023.
[2] Dunn PL, North RJ. Virulence ranking of some Mycobacterium tuberculosis and Mycobacterium bovis strains according to their ability to multiply in the lungs, induce lung pathology, and cause mortality in mice. Infect Immun. 1995 Sep;63(9):3428-37.
[3] Harris SA, Meyer J, Satti I et al. Evaluation of a human BCG challenge model to assess anti-mycobacterial immunity induced by BCG and a candidate TB vaccine, MVA85A, alone and in combination. J Infect Dis 2013; 209:1259-68.
[4] Minhinnick A, Harris S, Wilkie M, Peter J, Stockdale L, Manjaly-Thomas ZR, Vermaak S, Satti I, Moss P, McShane H. Optimization of a Human Bacille Calmette-Guerin Challenge Model: A Tool to Evaluate Antimycobacterial Immunity. J Infect Dis. 2016 Mar 1;213(5):824-30.
[5] Carter E, Morton B, ElSafadi D, Jambo K, Kenny-Nyazika T, Hyder-Wright A, Chiwala G, Chikaonda T, Chirwa AE, Gonzalez Sanchez J, Yip V, Biagini G, Pennington SH, Saunderson P, Farrar M, Myerscough C, Collins AM, Gordon SB, Ferreira DM. A feasibility study of controlled human infection with intradermal Bacillus Calmette-Guerin (BCG) injection: Pilot BCG controlled human infection model. Wellcome Open Res. 2024 Jun 5; 8:424.
[6] Krishnan N, Priestman M, Uhía I, Charitakis N, Glegola-Madejska IT, Baer TM, Tranberg A, Faraj A, Simonsson US, Robertson BD. A noninvasive BCG skin challenge model for assessing tuberculosis vaccine efficacy. PLoS Biol. 2024 Aug 19;22(8): e3002766.
[7] Medina, E., & North, R. J. (1998). Resistance ranking of some common inbred mouse strains to Mycobacterium tuberculosis and relationship to major histocompatibility complex haplotype and Nramp1 genotype. Immunology, 93(2), 270-274.
[8] Chackerian AA, Behar SM. Susceptibility to Mycobacterium tuberculosis: lessons from inbred strains of mice. Tuberculosis (Edinb). 2003;83(5):279-85.
[9] Dunn PL, North RJ. Virulence ranking of some Mycobacterium tuberculosis and Mycobacterium bovis strains according to their ability to multiply in the lungs, induce lung pathology, and cause mortality in mice. Infect Immun. 1995 Sep;63(9):3428-37.
[10] Lyadova IV, Eruslanov EB, Khaidukov SV, Yeremeev VV, Majorov KB, Pichugin AV, Nikonenko BV, Kondratieva TK, Apt AS. Comparative analysis of T lymphocytes recovered from the lungs of mice genetically susceptible, resistant, and hyperresistant to Mycobacterium tuberculosis-triggered disease. J Immunol. 2000 Nov 15;165(10):5921-31.
In the introduction, we add the scientific basis for the preparation of TB models by injecting BCG strains into the tail vein, and explain in detail its advantages in vaccine evaluation (Page 4, line 169). Additional content and references are as follows:
“The establishment of stable and reliable animal infection models is essential for the evaluation of the efficacy of TB vaccines. However, M. tb is highly infectious and pathogenic, and its manipulation must be carried out in a biosafety level 3 laboratory (BSL-3) environment, which involves high protection cost and high experimental risk [35]. BCG, a live attenuated vaccine strain of Mycobacterium bovis, has a high degree of genetic similarity to M. tb, but its virulence is markedly reduced, making it almost non-pathogenic to humans (except for immunocompromised individuals). Its study only needs to be conducted in a biosafety level 2 (BSL-2) laboratory, an environment that can significantly reduce costs and risks [35]. In recent years, alternative BCG models have been used in the field of TB vaccine development, with a number of studies confirming that a BCG-based human challenge model can quickly and cost-effectively detect differences in vaccine-induced immune responses, providing an efficient and safe evaluation method for early clinical screening of vaccine candidates [36]. Krishnan N et al. [37] also successfully established a murine BCG skin infection model and demonstrated its applicability for vaccine efficacy evaluation. Furthermore, the use of BALB/c mice, which have a uniform genetic background, has become a widely accepted animal model for TB vaccine research. This is mainly due to their ability to consistently repro-duce the pathological features of TB infection and to discriminate the differential protective efficacy of vaccines [10, 11, 38-41]. The present study is an innovative construction of a murine systemic infection model using intravenous injection of BCG, in order to simulate the infection process caused by hematogenous dissemination of M. tb. The aim of this study is to establish a safe and controllable platform for tuberculosis vaccine screening. This strategy can be used to rapidly screen potential vaccine candidates using the BCG model, which can be further verified by an M. tb infection experiment. The result of this is a significant improvement in the efficiency of TB vaccine development and a reduction in the cost of early research.”
[10] Liang Y, Wu X, Zhang J, et al. Immunogenicity and therapeutic effects of Ag85A/B chimeric DNA vaccine in mice infected with Mycobacterium tuberculosis [J]. FEMS Immunol Med Microbiol, 2012, 66(3): 419-26.
[11] Liang Y, Zhao Y, Bai X, et al. Immunotherapeutic effects of Mycobacterium tuberculosis rv3407 DNA vaccine in mice [J]. Autoimmunity, 2018, 51(8): 417-22.
[35] National Health Commission of the People's Republic of China. List of pathogenic microorganisms that are transmitted between humans. 2023.
[36] Minhinnick A, Harris S, Wilkie M, et al. Optimization of a Human Bacille Calmette-Guérin Challenge Model: A Tool to Evaluate Antimycobacterial Immunity [J]. J Infect Dis, 2016, 213(5): 824-30.
[37] Krishnan N, Priestman M, Uhía I, et al. A noninvasive BCG skin challenge model for assessing tuberculosis vaccine efficacy [J]. PLoS Biol, 2024, 22(8): e3002766.
[38] Lyadova I V, Eruslanov E B, Khaidukov S V, et al. Comparative analysis of T lymphocytes recovered from the lungs of mice genetically susceptible, resistant, and hyperresistant to Mycobacterium tuberculosis-triggered disease [J]. J Immunol, 2000, 165(10): 5921-31.
[39] Medina E, North R J. Genetically susceptible mice remain proportionally more susceptible to tuberculosis after vaccination [J]. Immunology, 1999, 96(1): 16-21.
[40] Christy A J, Dharman K, Dhandapaani G, et al. Epitope based recombinant BCG vaccine elicits specific Th1 polarized immune responses in BALB/c mice [J]. Vaccine, 2012, 30(7): 1364-70.
[41] Dobakhti F, Naghibi T, Taghikhani M, et al. Adjuvanticity effect of sodium alginate on subcutaneously injected BCG in BALB/c mice [J]. Microbes Infect, 2009, 11(2): 296-301.
- Comment: Line 154: The number of groups used was five.
Responses: Thank you for your careful attention, I have changed the "6 groups" to "5 groups". (Page 5, line 244)
- Comment: Line 197: It is necessary to clarify under which conditions one stimulus or another was added. It is unclear if the induction was by Ag or PHA.
Responses: Thank you very much for your valuable suggestion. We have revised the methods section of the paper to more clearly describe the experimental conditions and stimuli used (Page 7, line 301). Specific modifications are as follows:
“For vaccine immunogenicity experiments, 6 mice were sacrificed at 6 weeks after the last immunization. Pulmonary and splenic lymphocytes were isolated and seeded into 96-well cell culture plates at the concentrations of 1×105 cells/ml and 5×105 cells/ml, respectively, at 100 μl per well. Cells were then stimulated with three conditions: (1) 50 μl complete cell culture medium (90% RPMI-1640 + 10% FBS) as negative control; (2) 90 µg/ml PHA as positive control; and (3) 90 µg/ml Ag85AB recombinant protein as experimental treatment. The cell culture plate was placed in a humidified incubator at 37℃ with 5% COâ‚‚ for 48 hours. After 48 hours of culture, the splenic lymphocytes were placed in a 1.5ml centrifuge tube and centrifuged at 5000rpm for 3 minutes. The resulting supernatants were carefully collected and stored at -80℃ for subsequent analysis.”
- Comment: It is necessary to describe how the lymphocyte culture was carried out for supernatant collection. Additionally, an explanation should be provided as to why cytokines were not evaluated in situ, from cell lysates or plasma.
Responses: Thank you very much for your valuable suggestion. Given the limited volume of mouse peripheral blood samples, which have been used preferentially for the measurement of serum antibody levels, the remaining volume was insufficient for cytokine quantification. Therefore, in this study, the culture supernatants of spleen lymphocytes stimulated with Ag85AB specific antigen were collected to systematically evaluate the levels of cytokines related to vaccine-induced cellular immune response.
The baseline cytokine secretion levels of spleen lymphocytes not activated by specific antigens are very low, which cannot effectively characterize the strength of T cell immune response stimulated by vaccines.
In this study, we focused on the secreted extracellular cytokines (such as IFN-γ and IL-2, etc.) after antigen stimulation, rather than the intracellular storage factors. Therefore, we did not perform the cell lysis step, which can not only avoid the risk of protein degradation caused by lysis, but also can truly reflect the functional cellular immune response.
We have revised the Methods section of the paper to more clearly describe the lymphocyte culture and supernatant collection methods (Page 7, line 301). Specific modifications are as follows:
“For vaccine immunogenicity experiments, 6 mice were sacrificed at 6 weeks after the last immunization. Pulmonary and splenic lymphocytes were isolated and seeded into 96-well cell culture plates at the concentrations of 1×105 cells/ml and 5×105 cells/ml, respectively, at 100 μl per well. Cells were then stimulated with three conditions: (1) 50 μl complete cell culture medium (90% RPMI-1640 + 10% FBS) as negative control; (2) 90 µg/ml PHA as positive control; and (3) 90 µg/ml Ag85AB recombinant protein as experimental treatment. The cell culture plate was placed in a humidified incubator at 37℃ with 5% COâ‚‚ for 48 hours. After 48 hours of culture, the splenic lymphocytes were placed in a 1.5ml centrifuge tube and centrifuged at 5000rpm for 3 minutes. The resulting supernatants were carefully collected and stored at -80℃ for subsequent analysis.”
- Comment: Why were the experiments conducted with M. bovis? It seems important to conduct experiments with a more representative strain of the infection, such as a virulent strain.
Responses: Thank you very much for your valuable suggestion. We have given a detailed reply in comment 4 on the reasons why BCG strains were used to replace Mycobacterium tuberculosis in preparation of mouse tuberculosis model for preliminary evaluation of tuberculosis vaccines in this study, and the scientific basis for the preparation of tuberculosis model by injecting BCG strains through tail vein was added in the introduction section.
- Comment: It would be useful to include a brief description of the vaccine creation process.
Responses: Thank you very much for your suggestion. We have supplemented the description of the vaccine preparation process in the Methods section of the manuscript as follows (Page 5, line 235):
“Briefly, the DNA encoding amino acids 125-282 of the MTB H37Rv Ag85B protein was amplified by PCR using specific oligonucleotide primers containing the recognition site of the restriction enzyme Acc I. The purified PCR product was inserted into nucleotides 430-435 (the Acc I site) of the ag85a gene, and then cloned into the eukaryotic expression vector pVAX1 to construct a MTB ag85a/b chimeric DNA vaccine.”
Results:
- Comment: Make sure the term is homogenized throughout the text to maintain consistency. Figure 1 (PXAX1/ PVAX1)
Responses: Thank you for your careful observation. I am very sorry for this error, which is due to my negligence. I have changed "PXAX1" to "pVAX1" in the figure 1 and re-uploaded it.
- Comment: What tests were used? It is important to note that some conditions were performed with different sample sizes (n), which could bias the results (see figure 5).
Responses: Thank you very much for your suggestion.
We have provided a more detailed description of the experimental detection methods in the Methods section, including the supplementation of sample sizes and clarification of the statistical tests used. Specifically, the sample sizes were explicitly stated alongside the statistical outcomes in the Results section. In the Discussion section, we have added a discussion on the potential bias caused by differences in sample sizes in Figure 5. The specific modifications are as follows (Page 16, line 521):
“However, it is important to note that the results may not fully reflect the true biological differences due to inconsistent sample sizes between groups.”
- Comment: Inconsistencies in the number of samples analyzed are observed (figure 7A), as the methodology mentions the sacrifice of 4 or 6 mice.
Responses: Thank you very much for your suggestion. We have reviewed the data and method and have corrected any inconsistencies in the number of samples analyzed. The revised method and results sections reflect the correct number of mice sacrificed and analyzed.
We are deeply grateful for your comprehensive review, which has significantly enhanced the clarity and scientific rigor of our manuscript. We hope that these revisions address the reviewers' concerns and provide a more thorough evaluation of our manuscript. If there be any need for further improvements, please do not hesitate to inform us.
Best regards,
Authors: Zhen Zhang, Yan Liang and Xueqiong Wu
Reviewer 2 Report
Comments and Suggestions for Authors
Authors described the immunogenicity and immunotherapeutic efficacy of a Mtb 85ab DNA vaccine and compare the administration route through pulmonary and electroporation routes.
The manuscript is well designed and nicely presented. I have some comments.
Comments
- In page 3 line 137 check the concentration of bovis BCG preparation.
- In page 3 lines 143 and 144 check the phrase “In this study, one mouse …”
- Why FoxP3 T cells were not identified in lung?
- Define the CD4 and CD25 fluorochrome used and the FoxP3-PE reference.
- In Figure 1 correct pVAX1 name identify the A and B figures.
- Why are the Y axes of Figure 2A and B not the same?
- Why were pulmonary IFN-gamma secreting lymphocytes not quantified at 12 weeks post vaccination (Figure 2)? Justify in the manuscript because it is your main goal to compare the effective induction of a pulmonary immune response induced by a pulmonary delivered DNA vaccine.
- Use the cytokines abbreviation on page 5 instead on page 8.
- Verify the p values described on page 9 line 314. There are six cytokines in Ag85ab-PD group being compared to three other groups. The represented p values are for ?
- Verify in Figure 3D (GM-CSF) the asterisk for the statistical significance between ag85ab-EP and ag85ab-PD groups, since in the manuscript there is a statistical difference between both groups.
Author Response
We sincerely appreciate your positive evaluation of our manuscript and your recognition of the significance of our research. We are also very grateful for your valuable comments and suggestions. Your insightful review has greatly contributed to improving the clarity and scientific rigor of our work. We have carefully revised the manuscript to address your concerns. Below are our point-by-point responses to your suggestions.
Response to Reviewer 2’s comments:
- Comment: In page 3 line 137 check the concentration of bovis BCG preparation.
Responses: I am very sorry for the calculation error and have corrected the preparation concentration of the BCG suspension in the Methods section (Page 5, line 221).
- Comment: In page 3 lines 143 and 144 check the phrase “In this study, one mouse …”
Responses: Thank you for your careful reading. Following your suggestion, we have added "In this study, one mouse..." to the text. Corrected to "In this study, each mouse..." (Page 5, line 220)
- Comment: Why FoxP3 T cells were not identified in lung?
Responses: Thank you for your suggestion. Due to the small number of lung cells obtained during the treatment of lung tissue, only enough for ELISPOT to determine the number of efficient T lymphocytes secreting IFN-γ in lymphocytes and the cytokines in the supernatant of lung cell culture, the proportion of lung FoxP3 Tregs cells was not performed.
- Comment: Define the CD4 and CD25 fluorochrome used and the FoxP3-PE reference.
Responses: Thank you for your suggestion. We apologize for the oversight. The CD4 and CD25 fluorochromes used were CD4-FITC and CD25-APC, respectively. The FoxP3-PE antibody was used for intracellular staining. These details have been added to the manuscript for clarity. (Page 7, line 322)
- Comment: In Figure 1 correct pVAX1 name identify the A and B figures.
Responses: Thank you for your suggestion. We apologize for the oversight. We have corrected the labels for the A and B figures in Figure 1 to accurately reflect the pVAX1 vector used in our study.
- Comment: Why are the Y axes of Figure 2A and B not the same?
Responses: We sincerely thank the reviewers. The rationale behind the discrepancy in the nomenclature of the Y axes in Figures 2A and B is that Figure 2A corresponds to the number of spots secreted by lung lymphocytes, whereas Figure 2B corresponds to the number of spots secreted by spleen lymphocytes.
- Comment: Why were pulmonary IFN-gamma secreting lymphocytes not quantified at 12 weeks post vaccination (Figure 2)? Justify in the manuscript because it is your main goal to compare the effective induction of a pulmonary immune response induced by a pulmonary delivered DNA vaccine.
Responses: Thank you for your attention and valuable comments on our study. We regret to inform you that we were unable to ascertain the presence of IFN-γ-secreting lymphocytes in the lungs at 12 weeks after vaccination. This was due to errors made by the experimentalists during the operation at that time, which resulted in the relevant data not being accurately presented. We have added the following in the Methods section:
The sentence “Unfortunately, the number of IFN-γ-secreting effector T lymphocyte spots in the lungs at 12 weeks after the last immunization was lost due to operator error.” was inserted into lines. (Page 6, line 275)
- Comment: Use the cytokines abbreviation on page 5 instead on page 8.
Responses: Thank you for your suggestion. In light of the observations you have made, a thorough review of the complete text has been conducted, resulting in the consistent utilization of abbreviated forms of cytokines on page 5 and in subsequent sections. This approach was adopted to ensure uniformity in the terminology employed.
- Comment: Verify the p values described on page 9 line 314. There are six cytokines in Ag85ab-PD group being compared to three other groups. The represented p values are for ?
Responses: Thank you for your suggestion. We have reviewed the statistical analysis and validated the p-values presented in Figures 3A-3G and on line 314, page 9. The p-values indicate significant differences between the ag85ab+PD group and the other three groups for six Th1 cytokines. Due to the large number of comparisons resulting from multiple groups and cytokines (as detailed in Figures 3A-3G), these p-values have not been reported individually in the main text. We apologize for the lack of clarity in our initial presentation and have revised these statements uniformly to "all P<0.05". (Page 11, line 442)
- Comment: Verify in Figure 3D (GM-CSF) the asterisk for the statistical significance between ag85ab-EP and ag85ab-PD groups, since in the manuscript there is a statistical difference between both groups.
Responses:
Thank you for your suggestion. A thorough examination of Figure 3D and the accompanying text reveals no substantial disparities between ag85ab EP and ag85ab PD. Consequently, no asterisk is warranted. It is noteworthy that the present study does not elaborate on any substantial discrepancies between the two groups, a detail that was omitted in the original paper.
We are deeply grateful for your comprehensive review, which has significantly enhanced the clarity and scientific rigor of our manuscript. We hope that these revisions address the reviewers' concerns and provide a more thorough evaluation of our manuscript. If there be any need for further improvements, please do not hesitate to inform us.
Best regards,
Authors: Zhen Zhang, Yan Liang and Xueqiong Wu
Reviewer 3 Report
Comments and Suggestions for Authors
This manuscript compares the immunogenicity of the ag85ab DNA vaccine according to the immunization routes (electroporated intramuscular injection and pulmonary delivery) in preventive and therapeutic mouse models. The pulmonary delivery route induced earlier and more significant Th1 responses in the lung than intramuscular injection, but no significant difference in the spleen based on the ELISPOT assay and cytokines produced from antigen-restimulated cells. In the therapeutic model, the pulmonary route had a better effect than the intramuscular route based on ELISPOT assay of spleen cells, CFU count at 10 weeks after final immunization, and the pathologic findings at 4 weeks after final immunization. This manuscript shows new findings or advantages of the pulmonary route in terms of protection against tuberculosis. However, my main concerns are as follows:
1. In the vaccine field, strong immunogenicity is not always correlated with protective efficacy. Although the pulmonary delivery route is expected to produce more good efficacy, authors must perform a challenge test with Mtb to solidly confirm that PD route is best.
2. In the preventive models, the assay was performed at 4 weeks and 10 weeks after the final immunization. But, authors should consider performing at 10 days after the second and third immunization.
3. Please determine multiple or triple cytokine-producing T cells (example: IFN-r/IL-2/IL-17 or IFN-r/TNF-alpha/IL-17) which are correlated with protective efficacy.
4. In Figures 2, 3, and 4, 90 ug/ml to 30 ug/ml of Ag85AB protein was used to restimulate lung and spleen cells. However, this concentration is too high because memory cells are usually stimulated with 2 to 10 ug/ml.
5. In the therapeutic model, vaccination was started at 3 days after infection, but it is needed to time of about 2~3 weeks to multiply the bacteria in the lung, and then immunized with a therapeutic vaccine. It is much better to determine adjunctive efficacy between anti-tuberculosis agents and a therapeutic vaccine. Because there is a limit to controlling tuberculosis only with therapeutic vaccines.
6. In the Figure 8, the immune assay results in the lung cells are absent.
Author Response
We sincerely appreciate your positive evaluation of our manuscript and your recognition of the significance of our research. We are also very grateful for your valuable comments and suggestions. Your insightful review has greatly contributed to improving the clarity and scientific rigor of our work. We have carefully revised the manuscript to address your concerns. Below are our point-by-point responses to your suggestions.
Response to Reviewer 3’s comments:
- Comment: In the vaccine field, strong immunogenicity is not always correlated with protective efficacy. Although the pulmonary delivery route is expected to produce more good efficacy, authors must perform a challenge test with M. tb to solidly confirm that PD route is best.
Responses: We fully agree that strong immunogenicity does not necessarily correlate with protective efficacy, and that the MTB challenge test remains the gold standard for evaluating vaccine efficacy. Therefore, this study not only focuses on enhancing local pulmonary immune responses induced by pulmonary delivery of ag85ab DNA vaccine, but also provides preliminary validation of its immunotherapeutic effects in a murine BCG infection model.
Regarding the scientific basis for using BCG strains as a surrogate for MTB in the challenge test to preliminarily evaluate vaccine efficacy, we have addressed this in the comment 4 of Reviewer 1 and added the scientific basis for establishing the infection model by tail vein injection of BCG strain in the Introduction section. We have revised the conclusion to explicitly state: “In the future, the M. tb challenge trial will be used to further validate the efficacy of the pulmonary delivery of the ag85ab DNA vaccine.” (Page 23, line 826)
The contents and references appended to the introduction are enumerated herewith (Page 4, line 169):
“The establishment of stable and reliable animal infection models is essential for the evaluation of the efficacy of TB vaccines. However, M. tb is highly infectious and pathogenic, and its manipulation must be carried out in a biosafety level 3 laboratory (BSL-3) environment, which involves high protection cost and high experimental risk [35]. BCG, a live attenuated vaccine strain of Mycobacterium bovis, has a high degree of genetic similarity to M. tb, but its virulence is markedly reduced, making it almost non-pathogenic to humans (except for immunocompromised individuals). Its study only needs to be conducted in a biosafety level 2 (BSL-2) laboratory, an environment that can significantly reduce costs and risks [35]. In recent years, alternative BCG models have been used in the field of TB vaccine development, with a number of studies confirming that a BCG-based human challenge model can quickly and cost-effectively detect differences in vaccine-induced immune responses, providing an efficient and safe evaluation method for early clinical screening of vaccine candidates [36]. Krishnan N et al. [37] also successfully established a murine BCG skin infection model and demonstrated its applicability for vaccine efficacy evaluation. Furthermore, the use of BALB/c mice, which have a uniform genetic background, has become a widely accepted animal model for TB vaccine research. This is mainly due to their ability to consistently repro-duce the pathological features of TB infection and to discriminate the differential protective efficacy of vaccines [10, 11, 38-41]. The present study is an innovative construction of a murine systemic infection model using intravenous injection of BCG, in order to simulate the infection process caused by hematogenous dissemination of M. tb. The aim of this study is to establish a safe and controllable platform for tuberculosis vaccine screening. This strategy can be used to rapidly screen potential vaccine candidates using the BCG model, which can be further verified by an M. tb infection experiment. The result of this is a significant improvement in the efficiency of TB vaccine development and a reduction in the cost of early research.”
[10] Liang Y, Wu X, Zhang J, et al. Immunogenicity and therapeutic effects of Ag85A/B chimeric DNA vaccine in mice infected with Mycobacterium tuberculosis [J]. FEMS Immunol Med Microbiol, 2012, 66(3): 419-26.
[11] Liang Y, Zhao Y, Bai X, et al. Immunotherapeutic effects of Mycobacterium tuberculosis rv3407 DNA vaccine in mice [J]. Autoimmunity, 2018, 51(8): 417-22.
[35] National Health Commission of the People's Republic of China. List of pathogenic microorganisms that are transmitted between humans. 2023.
[36] Minhinnick A, Harris S, Wilkie M, et al. Optimization of a Human Bacille Calmette-Guérin Challenge Model: A Tool to Evaluate Antimycobacterial Immunity [J]. J Infect Dis, 2016, 213(5): 824-30.
[37] Krishnan N, Priestman M, Uhía I, et al. A noninvasive BCG skin challenge model for assessing tuberculosis vaccine efficacy [J]. PLoS Biol, 2024, 22(8): e3002766.
[38] Lyadova I V, Eruslanov E B, Khaidukov S V, et al. Comparative analysis of T lymphocytes recovered from the lungs of mice genetically susceptible, resistant, and hyperresistant to Mycobacterium tuberculosis-triggered disease [J]. J Immunol, 2000, 165(10): 5921-31.
[39] Medina E, North R J. Genetically susceptible mice remain proportionally more susceptible to tuberculosis after vaccination [J]. Immunology, 1999, 96(1): 16-21.
[40] Christy A J, Dharman K, Dhandapaani G, et al. Epitope based recombinant BCG vaccine elicits specific Th1 polarized immune responses in BALB/c mice [J]. Vaccine, 2012, 30(7): 1364-70.
[41] Dobakhti F, Naghibi T, Taghikhani M, et al. Adjuvanticity effect of sodium alginate on subcutaneously injected BCG in BALB/c mice [J]. Microbes Infect, 2009, 11(2): 296-301.
The conclusion has been revised as follows (Page 23, line 826).:
In conclusion, M. tb ag85ab DNA vaccine delivered by both the pulmonary delivery route and the intramuscular electroporation route were able to effectively induce IFN-γ-secreting effector T lymphocyte responses in the lung and spleen, and induced a predominance of Th1- and Th17-type cellular immune responses. However, the DNA vaccine delivered by the pulmonary delivery route was able to induce more pulmonary effector T cells and a stronger response, induced higher levels of specific antibody IgA production in the respiratory mucosa, and showed promising efficacy in treating a mouse BCG infection model, whereas the intramuscular electroporation route induced higher levels of specific antibodies IgG, IgG1, and IgG2α in plasma. Therefore, the DNA vaccine immunized by pulmonary delivery can more effectively stimulate the body to produce stronger cellular immunity and mucosal immunity than intramuscular electroporation, especially local cellular immunity in the lung, which can significantly enhance the immunogenicity and therapeutic efficacy of M. tb ag85ab DNA vaccine. In the future, the M. tb challenge trial will be used to further validate the efficacy of the pulmonary delivery of the ag85ab DNA vaccine.
- Comment: In the preventive models, the assay was performed at 4 weeks and 10 weeks after the final immunization. But authors should consider performing at 10 days after the second and third immunization.
Responses: Thank you for your suggestion. We recognize the significant value of additional testing at 10 days after the second and third immunizations, which will help us to observe the dynamic changes in the short-term immune response. However, the main objective of this study is to evaluate the therapeutic efficacy of the vaccine in the treatment of tuberculosis, while comparing the long-term immune responses of the two immunization routes. Considering that it may be difficult to achieve satisfactory treatment outcomes if the number of immunotherapy sessions is too small, the detection time points are set at 4 weeks and 10 weeks after the last immunization. In the future, we plan to add more time points to provide a more comprehensive and in-depth analysis of immunokinetic.
- Comment: Please determine multiple or triple cytokine-producing T cells (example: IFN-r/IL-2/IL-17 or IFN-r/TNF-alpha/IL-17) which are correlated with protective efficacy.
Responses: Thank you for your suggestion. The role of multifunctional T cells in immune protection against TB has been extensively studied, and we fully recognize the scientific importances of such analytical approaches. However, this study mainly used ELISPOT to quantify T cells secreting a single cytokine (IFN-γ) without further quantification of T cells producing triple cytokines. Notably, we compensated for this by conducting Luminex multiplex assays to profile broad-spectrum cytokine levels (including Th1/Th2/Th17-associated cytokines). In the future, we will use flow cytometry to detect T cells co-expressing IFN-γ, IL-2, IL-17, or TNF-α. This will allow us to further analyze the protective efficacy of the pulmonary delivery route, and gain a more in-depth understanding of the complexity of the immune response and the protective mechanism.
- Comment: In Figures 2, 3, and 4, 90 ug/ml to 30 ug/ml of Ag85AB protein was used to restimulate lung and spleen cells. However, this concentration is too high because memory cells are usually stimulated with 2 to 10 ug/ml.
Responses: We sincerely appreciate your insightful suggestion. In our previous experiments, we used a recombinant Ag85AB protein at a final concentration of 30 μg/ml (stock solution 90 μg/ml) for stimulation. Through our systematic testing, we found that the number of effector T cells secreting IFN-γ peaked when the final concentration of Ag85AB protein for stimulation was between 20 and 30 μg/ml. However, our dose-response analysis identified 20-30 μg/ml as the optimal range for maximizing the number of IFN-γ-secreting effector T cell spots, and we recognize that these supraphysiological levels of stimulation may inadvertently mask authentic recall responses from antigen-experienced memory T cell subsets. To better simulate physiological antigen exposure conditions, a refined stimulation protocol using 2-10 µg/ml of Ag85AB protein will be implemented in future experiments. This will ensure the authenticity of the research when analyzing antigen-stimulated memory cells.
- Comment: In the therapeutic model, vaccination was started at 3 days after infection, but it is needed to time of about 2~3 weeks to multiply the bacteria in the lung, and then immunized with a therapeutic vaccine. It is much better to determine adjunctive efficacy between anti-tuberculosis agents and a therapeutic vaccine. Because there is a limit to controlling tuberculosis only with therapeutic vaccines.
Responses: Thank you for your suggestion. We fully agree that it takes time for MTB to multiply and that vaccination 3 days after infection may not fully mimic the clinical situation. It is also recognized that it may be more clinically relevant to evaluate the efficacy of therapeutic vaccines as adjuvant chemotherapy. We chose to administer the vaccine alone (without chemotherapy) 3 days after infection with the BCG strain for the following reasons: (1) The BCG infection model has been successfully established: In this study, a mouse model of acute infection was established by injecting a large amount of BCG strain (10.36×106 CFUs) into the tail vein. The BCG strain spread rapidly through the bloodstream and was evenly distributed to organs throughout the body (such as the lungs, liver, spleen, etc.), and the lung colony count of the mice reached log 10 (4.03±0.90) CFUs at this time. (2) The vaccine takes time to take effect: Although the TB DNA vaccine is administered 3 days after infection, it usually takes 2-3 weeks for the vaccine to achieve full anti-TB immunity. During this time, the mice have developed disease. (3) Evaluation of vaccine efficacy: We need to design an evaluation system that can clearly distinguish the difference between the vaccine group and the control group to assess whether the vaccine has an anti-TB protective effect. The single vaccine intervention protocol could avoid the synergistic/interfering effect of chemotherapy drugs and ensure that the observed protective effect is directly attributable to the immune effect of the vaccine. (4) Vaccine has limited effect on severe tuberculosis mice: our previous study found that vaccine immunotherapy alone could not achieve significant effect on mice with severe disease and immunocompromised/immunodeficient state. (5) The significant effect of chemotherapy easily masked the effect of vaccine: the therapeutic effect of chemotherapy on the drug-sensitive MTB infection model was very good, and the effect of vaccine was difficult to highlight in the case of combined treatment. Therefore, in the past, we were able to observe the therapeutic effect of vaccine by using combination treatment with multi-drug resistant MTB infection model [1]. Therefore, this study adopted a hierarchical and progressive research strategy: first, the BCG acute infection model was used to complete the preliminary screening of vaccine candidates, and then, based on the positive results, the efficacy of the MTB infection model was further verified, and the synergistic mechanism of the vaccine and chemotherapy regimen was systematically evaluated.
[1] Liang Y, Wu X, Zhang J, Li N, Yu Q, Yang Y, Bai X, Liu C, Shi Y, Liu Q, Zhang P, Li Z.The treatment of mice infected with multi-drug-resistant Mycobacterium tuberculosis using DNA vaccines or in combination with rifampin. Vaccine. 2008 Aug 18;26(35):4536-40.
- Comment: In the Figure 8, the immune assay results in the lung cells are absent.
Responses: We would like to express our sincere regret for the omission of the immunoassay results of lung cells from Figure 8. At the 4-week and 10-week marks following the final immunotherapy administration, a total of five mice were euthanized. Given the limited sample size of five rats per group, the left lung was allocated for colony counting, while the right lung underwent histopathological examination. Consequently, the detection of lung cell immunity was not feasible in the present experiment due to the restricted sample size. In future experiments, we will enhance the pulmonary immune detection scheme and increase the number of experimental mice to ensure a comprehensive pulmonary immune evaluation.
We are deeply grateful for your comprehensive review, which has significantly enhanced the clarity and scientific rigor of our manuscript. We hope that these revisions address the reviewers' concerns and provide a more thorough evaluation of our manuscript. If there be any need for further improvements, please do not hesitate to inform us.
Best regards,
Authors: Zhen Zhang, Yan Liang and Xueqiong Wu
Round 2
Reviewer 1 Report
Comments and Suggestions for Authors
The comments have been addressed objectively, so I consider accepting it for publication
Author Response
Comment: The comments have been addressed objectively, so I consider accepting it for publication.
Responses:
We sincerely appreciate your insightful feedback and constructive comments, which have greatly improved the quality of our manuscript. We are grateful for your thorough evaluation and are honored by your recommendation for acceptance. Thank you for your time and expertise in helping us to revise and improve our manuscript. We look forward to contributing to the scientific community with this research.
Reviewer 3 Report
Comments and Suggestions for Authors
I think that a challenge test with Mycobacterium tuberculosis must be needed at least.
Round 3
Reviewer 3 Report
Comments and Suggestions for Authors
Thank you for your response. I understand your situation. But therapeutic model using BCG strain is additional data and main data is preventive effect in this study. So as I commented previously, The preventive efficacy of Ag85 DNA administered via the pulmonary route must be determined using BCG or Mtb H37Ra, even when the use of Mtb H37Rv is limited. Sorry for not giving a positive response.
Author Response
Comment: Thank you for your response. I understand your situation. But therapeutic model using BCG strain is additional data and main data is preventive effect in this study. So as I commented previously, The preventive efficacy of Ag85 DNA administered via the pulmonary route must be determined using BCG or Mtb H37Ra, even when the use of Mtb H37Rv is limited. Sorry for not giving a positive response.
Responses:
We sincerely appreciate your valuable and constructive suggestions. We have followed your and the editor's suggestion to remove the immunotherapy-related data from the manuscript, to retain the immunogenicity analysis section, and to provide an in-depth discussion of the pathogenesis mechanisms of M. tb as a pulmonary pathogen, and the implications of vaccine strategies targeting pulmonary immune responses. The specific changes were implemented as follows (Page 17, line 534):
“M. tb usually infects the lungs via the respiratory tract. It is extremely cunning, able to evade the body's immune response by various means and successfully parasitize within host macrophages [42]. Its unique cell wall composition allows it to resist the bactericidal effects of macrophages. In addition, M. tb has the ability to evade the host immune system by secreting specific proteins that inhibit the bactericidal function of macrophages. Following infection, M. tb also induces a strong inflammatory response, resulting in the release of large numbers of inflammatory cells and cytokines. However, this response is often insufficient to eradicate the pathogen and can lead to tissue damage. Furthermore, M. tb has the ability to enter a latent state in the host, where it can persist for long periods of time. In immunocompromised individuals, this latent pathogen can be reactivated and cause disease manifestations [43]. Selection of the appropriate vaccination method and route has been shown to be one of the important factors in inducing effective immune responses to TB vaccine [44]. It is imperative to emphasize the importance of targeting the pulmonary immune response in the development of vaccines against M. tb. The lung is the primary site of infection for M. tb, and vaccines targeting the lung can elicit a local immune response and enhance lung defensive. For example, studies have shown that inducing an immune response in the lung mucosa to produce specific IgA antibodies can prevent the attachment and invasion of pathogens at an early stage of invasion [45-48]. In addition, lung-targeted vaccines have been shown to promote the development of lung-resident memory T cells [49], which have the ability to persist in the lung for longer periods and trigger a rapid and effective immune response upon subsequent exposure to M. tb. TB vaccine candidates, such as PPE15-LMQ, have been shown to have the ability to elicit robust CD4+ T cell responses in the lungs, even after systemic administration. These responses exhibit a resident memory phenotype, which is associated with protective immunity against TB. This local immune response not only provides prolonged protection but also mitigates immunopathological damage [43]. Many studies on mucosal immunity have shown that multifunctional Th1 cells, which produce high levels of cytokines locally in the lung are more effective in protecting against TB than the systemic immune responses exhibited in the spleen [50-53].To enhance local lung immunity, this study used a pulmonary delivery device for the first time to deliver a DNA vaccine directly into the lungs of mice, and compared the characteristics of their systemic immunity and local lung immunity with DNA vaccine by intramuscular electroporation.”
[42] Ernst J D. Antigenic Variation and Immune Escape in the MTBC [J]. Adv Exp Med Biol, 2017, 1019: 171-90.
[43] Korompis M, De Voss C J, Li S, et al. Strong immune responses and robust protection following a novel protein in adjuvant tuberculosis vaccine candidate [J]. Sci Rep, 2025, 15(1): 1886.
[44] Setiabudiawan T P, Reurink R K, Hill P C, et al. Protection against tuberculosis by Bacillus Calmette-Guérin (BCG) vaccination: A historical perspective [J]. Med, 2022, 3(1): 6-24.
[45] Fröberg J, Diavatopoulos D A. Mucosal immunity to severe acute respiratory syndrome coronavirus 2 infection [J]. Curr Opin Infect Dis, 2021, 34(3): 181-6.
[46] Oh J E, Song E, Moriyama M, et al. Intranasal priming induces local lung-resident B cell populations that secrete protective mucosal antiviral IgA [J]. Sci Immunol, 2021, 6(66): eabj5129.
[47] Marking U, Bladh O, Havervall S, et al. 7-month duration of SARS-CoV-2 mucosal immunoglobulin-A responses and protection [J]. Lancet Infect Dis, 2023, 23(2): 150-2.
[48] Zhou X, Wu Y, Zhu Z, et al. Mucosal immune response in biology, disease prevention and treatment [J]. Signal Transduct Target Ther, 2025, 10(1): 7.
[49] Ko K H, Bae H S, Park J W, et al. A vaccine platform targeting lung-resident memory CD4(+) T-cells provides protection against heterosubtypic influenza infections in mice and ferrets [J]. Nat Commun, 2024, 15(1): 10368.
[50] Forbes E K, Sander C, Ronan E O, et al. Multifunctional, high-level cytokine-producing Th1 cells in the lung, but not spleen, correlate with protection against Mycobacterium tuberculosis aerosol challenge in mice [J]. J Immunol, 2008, 181(7): 4955-64.
[51] Jeyanathan M, Heriazon A, Xing Z. Airway luminal T cells: a newcomer on the stage of TB vaccination strategies [J]. Trends Immunol, 2010, 31(7): 247-52.
[52] Santosuosso M, Zhang X, McCormick S, et al. Mechanisms of mucosal and parenteral tuberculosis vaccinations: adenoviral-based mucosal immunization preferentially elicits sustained accumulation of immune protective CD4 and CD8 T cells within the airway lumen [J]. J Immunol, 2005, 174(12): 7986-94.
[53] Santosuosso M, McCormick S, Roediger E, et al. Mucosal luminal manipulation of T cell geography switches on protective efficacy by otherwise ineffective parenteral genetic immunization [J]. J Immunol, 2007, 178(4): 2387-95.
We sincerely appreciate the time and effort you have dedicated to advancing this manuscript. We hope these responses address the reviewers' concerns and contribute to a more comprehensive evaluation of our manuscript. If there be any need for further improvements, please do not hesitate to inform us.